# LLM-Assisted Code Cleaning For Training Accurate Code Generators

**Naman Jain, Tianjun Zhang, Wei-Lin Chiang, Joseph E. Gonzalez, Koushik Sen & Ion Stoica**
University of California, Berkeley
`{naman_jain,tianjunz,weichiang,jegonzal,ksen,istoica}@berkeley.edu`

## Abstract

Natural language to code generation is an important application area of LLMs and has received wide attention from the community. The majority of relevant studies have exclusively concentrated on increasing the quantity and functional correctness of training sets while disregarding other stylistic elements of programs. More recently, data quality has garnered a lot of interest and multiple works have showcased its importance for improving performance. In this work, we investigate data quality for code and find that making the code more structured and readable leads to improved code generation performance of the system. We build a novel data-cleaning pipeline that uses these principles to transform existing programs by 1.) renaming variables, 2.) modularizing and decomposing complex code into smaller helper sub-functions, and 3.) inserting natural-language based plans via LLM based transformations. We evaluate our approach on two challenging algorithmic code generation benchmarks and find that fine-tuning CodeLLaMa-7B on our transformed modularized programs improves the performance by up to **30%** compared to fine-tuning on the original dataset. Additionally, we demonstrate improved performance from using a smaller amount of higher-quality data, finding that a model fine-tuned on the entire original dataset is outperformed by a model trained on 15% of our cleaned dataset. Even in comparison to closed-source models, our models outperform the much larger AlphaCode models (Li et al., 2022)

## 1 Introduction

Natural language to code generation has witnessed considerable advances in recent years with the advent of large language models (LLMs for brevity). These advances primarily arise from training on large web-scale data and are measured based on the functional correctness of the programs. Thus, other aspects like readability, structuring, and styling and how they affect training and data quality are largely ignored by these works. On the flip side, many recent works have demonstrated the effectiveness of training on higher quality data during both pre-training (Li et al., 2023d) and fine-tuning (Zhou et al., 2023; Cao et al., 2023) phases. Even within the code-generation domain, Gunasekar et al. (2023) demonstrated the benefits of training on a "textbook" quality dataset, generated synthetically using the GPT-3.5-turbo model (Ouyang et al., 2022). However, these works do not provide an understanding of the factors that actually improve the data quality.

In this work, we show that using programs following good programming practices and allowing for more readability leads to improved code generation performance compared to using programs that do not follow these practices. We use these insights to build a novel automated code data-cleaning pipeline that transforms programs while maintaining functional correctness using input-output examples. In contrast to prior works that curate *high quality* datasets by directly generating *new* data using LLMs, here we translate existing datasets into their *parallel cleaned* versions while identifying attributes that actually improve data quality.

We use LLMs to perform the transformations used in our data-cleaning approach. We demonstrate that instruction-tuned models can take a user-identified attribute of data quality as a natural language instruction and perform the transformation accurately. Our approach leverages the disparity in difficulty between generating a solution and editing an existing one. Therefore, it is particularly effective in domains where the existing model struggles to generate a correct solution but can effectively edit

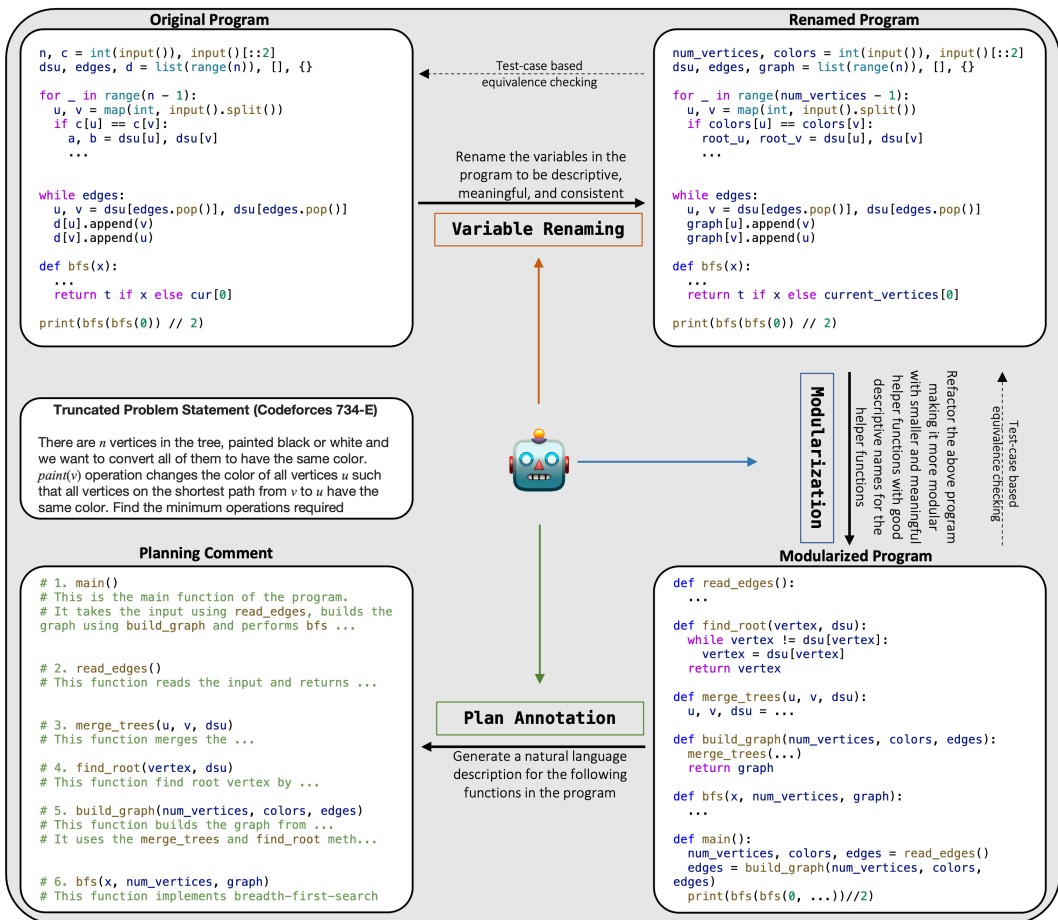

Figure 1: **The overview of our code cleaning approach**. We apply instruction-tuned LLMs to transform existing datasets by providing natural language prompts and use input-output examples to maintain function equivalence between original and transformed programs. Our cleaning approach works in three steps. The top-left figure depicts the original program from the dataset. This program first undergoes variable renaming (top-right figure). Next, the renamed program is decomposed into constituent sub-functions and converted into a modularized program (bottom-right figure). Finally, we generate a natural-language *plan* from the modularized program by summarizing the functions in a top-down manner (bottom-left figure). This plan is prepended to the program as a comment. The middle-left figure presents the truncated problem statement

a given solution. We perform our data-cleaning transformations in three iterations: 1) renaming variables 2) modularizing complex code into subfunctions, and 3) adding planning annotations.

Figure 1 provides an overview of our approach. Notice that the variable renaming step at the top adjusts the variable names to be contextually relevant (e.g. `a` to `root_u` and `d` to `graph`). The modularization step (depicted on the right) identifies and decomposes the original program into several smaller subfunctions such as `find_root`, `merge_trees`, `build_graph`, etc. It then implements these subroutines and assembles the modular program. Finally, our planning step (depicted at the bottom) constructs a plan by summarizing functions in a top-down fashion (starting from the `main`).

We evaluate our approach in a niche, yet challenging, domain of algorithmic code generation. The goal is to generate a program for a given problem statement. The task is challenging because it requires both high-level algorithmic reasoning and low-level coding and is evaluated using a strict functional correctness metric. We use two well-known algorithmic code generation benchmarks, namely APPS (Hendrycks et al., 2021) and CODE-CONTESTS (Li et al., 2022). We transform the corresponding programs in the training sets and obtain *parallel* datasets from our cleaning approach. Additionally, we utilize input-output examples to maintain functional equivalence between the original and transformed programs. We qualitatively analyze the generated dataset and find that it uses

smaller helper sub-functions, each often implementing a standard algorithm or key program functionality, and provide more in depth findings in Section 4.1. We further assess the impact of the transformed datasets on the performance on our downstream code generation task. We fine-tune CODELLAMA-7B model on the various collected datasets. Our findings reveal that the model fine-tuned on our modularized dataset outperforms the model fine-tuned on the functionally equivalent original dataset by up to **30%**. Beyond, performance improvement, we also demonstrate that improving data quality improves the data efficiency. In particular, a model fine-tuned on the entire original dataset is outperformed by a model trained on just 15% of our cleaned dataset.

We next study improving planning in a supervised learning setup similar to prior works (Fu et al., 2023; Li et al., 2023b). While we observe limited improvements in planning, we disentangle planning vs coding capabilities and find that our fine-tuned model is capable of using gold-annotated plans, extracted from the ground-truth solutions to accurately generate solutions for the complex programs. This highlights planning for complex problems remaining a key bottleneck that does not seem to improve by merely increasing training datasets. Finally, in comparison to existing baselines, our fine-tuned models outperform the larger ALPHACODE (Li et al., 2022) models.

## 2 METHODOLOGY

In this section, we present our general data transformation approach and then instantiate it for performing code data cleaning.

### 2.1 TRANSFORMATIONS FOR DATA CLEANING

Given a dataset $\mathcal{D}$ consisting of $N$ instances $\mathbf{d}_i$, such that, $\mathcal{D} = \{\mathbf{d}_i\}_{i=1}^N$. To achieve a desired data cleaning specification, the user additionally provides a data-cleaning instruction $\mathcal{I}$, which highlights an attribute that needs to be modified. Optionally, we also use an oracle equivalence checker ($\mathcal{O}$) which ensures that the transformed data instance $\tilde{\mathbf{d}}_i$ is consistent with the original input based on some desired metric. For example, we can use edit-distance or functional equivalence based on input-output examples as our oracle checker.

We use a pre-trained language model (denoted by $\mathcal{M}$) to generate the transformed instance ($\tilde{\mathbf{d}}_i$) by prompting the model with the transformation instruction ($\mathcal{I}$) and the original answer ($\mathbf{y}$). We can perform either zero-shot or few-shot prompting for performing the data cleaning operation. Finally, we extract the instance $\tilde{\mathbf{d}}_i$ generated by $\mathcal{M}$, and apply our oracle equivalence checker ($\mathcal{O}$) to ensure consistency with the original data. If $\mathcal{O}(\tilde{\mathbf{d}}_i, \mathbf{d}_i) = 0$, i.e., the oracle reports a failure, we reject the generated output and retry the example within a sampling budget.

While our transformation approach does not provide any guarantees about the quality of the performed transformation and relies on LLMs, we empirically observe that instruction-tuned LLMs can perform various unstructured data cleaning steps quite effectively. We provide a detailed analysis of the generated outputs for our algorithmic code generation setting in Section 4.1. Finally, in accordance with existing literature on prompting LLMs, we found that using simple and precise, low-level instructions improves the performance and accuracy of the models in performing the operations. Thus, for complex data cleaning operations (refactoring), we find improvements by breaking it down and performing multiple operations iteratively (renaming followed by modularization).

### 2.2 CODE DATA-CLEANING

We apply our transformations-based data cleaning approach to programming data. Coding requires both – low-level programming and high-level reasoning or planning skills. Therefore, we propose a three-step cleaning pipeline that improves the readability and program structuring targeting the low-level coding skills and inserts natural-language based plans data targeting the high-level reasoning skills. Our steps are detailed below.

1. **Rename variables.** This step renames the variables in the program, making them descriptive and easier to follow. Figure 1 top provides an example of this transformation.
2. **Modularize functions.** Problem decomposition has been identified as a key approach for improving the reasoning capabilities of models (Zhou et al., 2022; Wang et al., 2023). We

|  | split | APPS-INTRODUCTORY | APPS-INTERVIEW | APPS-COMPETITION | CODE-CONTESTS |
|---|---|---|---|---|---|
| Problems count | train | 42 | 1247 | 361 | 7132 |
|  | test | 702 | 2699 | 309 | 165 |
| Tests count | train | 1 | 1 | 9 | 200 |
|  | test | 10 | 19 | 39 | 200 |
| Solutions count | train | 736 | 18394 | 5060 | 98582 |

Table 1: Details about the number of problems, the median number of test cases per problem, and the number of solutions in the APPS and CODE-CONTESTS datasets.

> identify program decompositions and transform the program by extracting their functionality into smaller helper functions. Figure 1 right provides an example of this transformation.
>
> 3. **Plan annotations.** This step summarizes the helper functions in the already modularized program and prepends it to the programs in the form of a natural language plan. These natural language descriptions are analogous to prompting approaches that are used for solving reasoning problems like chain-of-thought prompting (Wei et al., 2022), parsel (Zelikman et al., 2023), etc. Figure 1 bottom provides an example of this transformation.

Additionally, while performing these transformations, we use the test cases provided in the dataset to construct our oracle equivalence checker ($\mathcal{O}$). It ensures that our transformed programs maintain functional equivalence to the original program.

## 3 EXPERIMENTAL SETUP

In this section, we detail our experimental setup and implementation. Section 3.1 outlines the benchmarks and metrics used for the algorithmic code generation task, while Sections 3.2 and 3.3 delve into the specifics of our code cleaning approach and fine-tuning experiments respectively.

### 3.1 BENCHMARKS

We use two standard algorithmic code generation benchmarks, APPS and CODE-CONTESTS. The benchmarks provide a collection of problem statements described in natural language and corresponding test cases. The goal is to generate a program that successfully solves the problem. The evaluation is performed using a strict functional correctness metric.

**APPS (Hendrycks et al., 2021).** This benchmark includes 10,000 problems, evenly split between training and test sets. It is sourced from multiple open-access competitive programming websites. It is further divided into APPS-INTRODUCTORY, APPS-INTERVIEW, and APPS-COMPETITION subsets based on problem difficulty. In this study, we only consider problems sourced from a subset of the competition websites based on the number of test cases provided.

**CODE-CONTESTS (Li et al., 2022).** This benchmark includes 13,328 problems in the training set and 165 problems in the test set. We only use a subset of the training split that includes `python` solutions satisfying the provided test cases. Additionally, since the training set provides over a hundred solutions per problem, we perform LSH based near-deduplication on the solutions and limit them to a maximum of 25 solutions per problem.

Table 1 and Appendix A provide further details about our final datasets.

**Metrics.** We assess the code generation performance of the models using the PASS@$K$ metric (Kulal et al., 2019; Chen et al., 2021), which evaluates the functional correctness of generated programs. For each problem, we generate $N$ solutions (where $N \geq 2K$) and compute the expected number of scenarios in which the problem is solved at least once when sub-selecting a random sample of $K$ solutions. We vary $K$ in $\{1, 10, 25\}$ for APPS dataset and $\{1, 10, 100\}$ for the CODE-CONTESTS benchmark. We present more details about sampling hyperparameters in Appendix A.

### 3.2 DATA TRANSFORMATIONS

We apply our data transformation approach on the APPS and CODE-CONTESTS datasets. Unless specified otherwise, we use GPT-3.5-TURBO as our default language model $\mathcal{M}$ to perform the transformations and use a default temperature 0.3. In case of failure, we retry up to 5 iterations. We

| Dataset | Notation | Applied On | Transformation Instruction ($\mathcal{I}$) |
|---|---|---|---|
| Base | $\mathcal{D}_{original}$ | - | - |
| Rename | $\mathcal{D}_{rename}$ | $\mathcal{D}_{original}$ | *Rename the variables in the program to be descriptive, meaningful, and consistent* |
| Modularize | $\mathcal{D}_{modular}$ | $\mathcal{D}_{rename}$ | *Refactor the above program making it more modular with smaller and meaningful helper functions with good descriptive names for the helper functions* |
| Plan | $\mathcal{D}_{planning}$ | $\mathcal{D}_{modular}$ | *Generate a natural language description for the following functions in the program* |

Table 2: Transformed datasets generated by our code cleaning approach. For each transformation, we have provided the corresponding notation, the transformation instruction used to perform the cleaning step and the dataset the transformation was applied on.

obtain three *parallel* datasets at the end of our cleaning process, one for each of renaming, modularization, and planning (note that the transformations are applied sequentially). Table 2 provides a summary of the generated datasets along with the instructions used to generate them. We provide complete details about the transformations in Appendix B.

We also simulate a simple direct synthetic data generation approach somewhat similar to Gunasekar et al. (2023). Specifically, we generate solutions for the training problems using the GPT-3.5-TURBO model. We use in-context learning with the two-shot prompt examples selected from our $\mathcal{D}_{modular}$ dataset. To ensure diverse solutions, we use three distinct few-shot examples and generate eight solutions for every prompt at a temperature of $0.5$. Additionally, we filter the solutions for correctness based on the ground truth test cases provided in the dataset to ensure we are not training on incorrect programs. Since it resembles a distillation-like setup, we refer to this dataset as $\mathcal{D}_{distill}$.

### 3.3 EXPERIMENT DETAILS

To evaluate the *quality* of the transformed datasets, we measure how they impact the test benchmark accuracy. We study both in-context learning and fine-tuning using examples from our datasets.

**Models.** We use the CODELLAMA-7B model (Rozière et al., 2023) in all our experiments (referred as CL-7B ahead). We use the model checkpoint from huggingface[1] and perform batched inference through vLLM (Kwon et al., 2023), necessary for computing the PASS@$K$ metric. We also present the numbers from CODE-DAVINCI-002 and GPT-3.5-TURBO whenever available.

**In-context learning.** We select two question-answer pairs from the $\mathcal{D}_{original}$ and $\mathcal{D}_{modular}$ training sets as our in-context learning example. For a fair comparison between the two evaluations, we use the same problem and corresponding solutions from the two datasets as examples. The examples are combined with appropriate delimiters and the model is then prompted with a new problem. Note that these in-context learning examples increase the sequence length by over 2,000 tokens and considerably slow the inference.

**Fine-Tuning.** We perform full fine-tuning over the base CL-7B model on the different datasets. We train the models for two epochs on the APPS dataset and one epoch on the CODE-CONTESTS dataset using a $5e^{-5}$ learning rate and an effective batch size of 256 on 4 A6000 GPUs.

## 4 EXPERIMENTAL RESULTS

We present our experimental results in this section. Section 4.1 first provides a qualitative overview of the transformed programs and Section 4.2 presents the main code generation results.

### 4.1 ANALYSIS OF THE TRANSFORMED PROGRAMS

**Data statistics.** For the CODE-CONTESTS dataset, out of 98,582 programs extracted from the original dataset ($\mathcal{D}_{original}$), we can successfully transform 92,675 (94.0%) into our modularized dataset ($\mathcal{D}_{modular}$). We obtain similar success rates for the APPS dataset (details deferred to the appendix). On the contrary, the distilled dataset ($\mathcal{D}_{distill}$), which is constructed by generating solutions directly using GPT-3.5-TURBO only finds a correct solution for about 50% of the problems.

**Analysis of the transformed programs.** We find that our transformation approach decomposes the original programs by inserting three new functions on a median (~2.6 functions on average).To get

---

[1]`https://huggingface.co/codellama/CodeLlama-7b-hf`
[2]Model generations were obtained from Chen et al. (2022a)

| | APPS-INTRODUCTORY | | | APPS-INTERVIEW | | |
|---|---|---|---|---|---|---|
| | PASS@1 | PASS@10 | PASS@25 | PASS@1 | PASS@10 | PASS@25 |
| **In-context Learning** | | | | | | |
| CL-7B + $\mathcal{D}_{original}$ | 14.2 | 29.2 | 38.4 | 1.8 | 7.3 | 10.4 |
| CL-7B + $\mathcal{D}_{modular}$ | 17.5 | 30.1 | 39.7 | 2.2 | 8.6 | 12.3 |
| | +3.3 | +0.9 | +1.3 | +0.4 | +1.3 | +1.9 |
| **Fine-tuning** | | | | | | |
| CL-7B + $\mathcal{D}_{original}$ | 18.7 | 34.4 | 40.2 | 3.4 | 9.7 | 13.6 |
| CL-7B + $\mathcal{D}_{modular}$ | **22.7** | **36.9** | 42.6 | **4.2** | **11.0** | **15.0** |
| | +4.0 | +2.5 | +2.4 | +0.8 | +1.3 | +1.4 |
| CL-7B + $\mathcal{D}_{planning}$ | 22.1 | **37.1** | 43.8 | 3.7 | 10.5 | 14.8 |
| CL-7B + $\mathcal{D}_{rename}$ | 19.2 | 36.6 | 42.9 | 4.0 | 10.7 | 14.6 |
| CL-7B + $\mathcal{D}_{distill}$ | 21.1 | 35.3 | 40.5 | 4.1 | 10.8 | 14.5 |
| **Closed models** | | | | | | |
| CODE-DAVINCI-002 [2] | 22.1 | 50.2 | 58.7 | 4.1 | 16.8 | 23.8 |

Table 3: **Results on APPS dataset.** We use the CODELLAMA-7B model (referred to as CL-7B) under in-context learning and fine-tuning. We use samples from the original and our transformed datasets and find that our cleaned datasets improve the performance of the model by over **20%**. The green highlighted numbers depict the improvements obtained from using $\mathcal{D}_{modular}$ (over $\mathcal{D}_{original}$). Similarly, using $\mathcal{D}_{rename}$ and $\mathcal{D}_{planning}$ also provide improvements, usually lesser than using $\mathcal{D}_{modular}$.

a better understanding of the decomposition, we cluster the functions using their function names and signatures. We find that these helper functions often implement key program logic, standard algorithms, and utilities like handling inputs, outputs, and orchestrating the main function. Interestingly, we also find that the helper functions are often reused across problems, with small variations in implementations. For example, the top five most frequent helper functions, `dfs`, `build_graph`, `gcd`, `dp`, and `binary_search` occur in about 3-8% of the problems. Additionally, we qualitatively analyze a hundred random samples from $\mathcal{D}_{original}$ and $\mathcal{D}_{modular}$ datasets to determine the quality of performed transformations. Figures 4 to 11 in the appendix provide examples of such transformations. We find that most of the transformations are meaningful. They improve the readability of the programs and also find suitable decomposition for the program logic encoded in the control flow (see Figures 4, 5, 6, 14 as examples). However, in some cases, the generated helper functions can have improper names (`calculate_max_colors` in Figure 11) or complex implementations copied directly from the original program (`count_sequences` in Figure 12). Additionally, for simpler programs (Figure 13), the entire program functionality can be implemented in a single function and the *decomposition* does not provide any extra information. Finally, we use GPT-4 as judge (Zheng et al., 2023) evaluation to quantitatively assess the transformations in regards to their meaningfulness and about the consistency of original and transformed programs. Appendix C.1 presents the comprehensive setup. We find that over 99% of the transformations are regarded as helpful of which only 3-5% of examples are judged as *can do better*. Similarly, 99.4% of the transformed programs are judged as consistent with the original programs. More detailed evaluation results in Table 6.

Unlike, generated code, we cannot constrain or check the generated natural language plans. Thus, we find that sometimes the plans can be imprecise and vary in detail. While using a stronger pretrained model like GPT-4 could alleviate some of these issues, we believe this will be a good avenue for applying something analogous to process supervision (Lightman et al., 2023).

## 4.2 MAIN RESULTS

Tables 3 and 4 provide our primary results on APPS and CODE-CONTESTS datasets respectively. We defer the results for the APPS-COMPETITION subset to Appendix C and highlight our findings below.

### 4.2.1 EFFECT OF MODULARIZATION

We find that our data-cleaning approach improves the performance of the model on both APPS and CODE-CONTESTS datasets in both in-context learning and fine-tuning settings.

---

[3] Result sourced from Li et al. (2022)

[4] Result sourced from Zhang et al. (2023b)

[5] Result sourced from Li et al. (2023c)

| | CODE-CONTESTS | | |
|---|---|---|---|
| | PASS@10 | PASS@25 | PASS@100 |
| **In-context Learning** | | | |
| CL-7B + $\mathcal{D}_{original}$ | 5.1 | 6.5 | 7.2 |
| CL-7B + $\mathcal{D}_{modular}$ | 4.9 | 6.6 | 9.3 |
| | -0.2 | +0.1 | +2.1 |
| **Fine-tuning** | | | |
| CL-7B + $\mathcal{D}_{original}$ | 5 | 6.4 | 10.9 |
| CL-7B + $\mathcal{D}_{modular}$ | **6.1** | **8.3** | 12.4 |
| | +1.1 | +1.9 | +1.5 |
| CL-7B + $\mathcal{D}_{planning}$ | 5.3 | 7.0 | 10.8 |
| CL-7B + $\mathcal{D}_{rename}$ | 4.7 | 6.3 | 10.5 |
| **Closed models** | | | |
| ALPHACODE-9B [3] | 5.0 | 7.0 | 10.0 |
| ALPHACODE-41B[3] | 5.0 | 7.0 | 10.0 |
| CODE-DAVINCI-002 [4] | 3.0 | - | 7.5 |
| GPT-3.5-TURBO[5] | - | - | 18.2 |
| + BRAINSTORM[5] | - | - | 29.3 |

Table 4: **Result on the CODE-CONTESTS dataset.** Similar to findings on the APPS dataset, we find that our data cleaning approach generally improves the performance with modularization working particularly well while planning and re-naming providing marginal to no improvements.

| | CODE-CONTESTS-PLAN | | |
|---|---|---|---|
| | PASS@10 | PASS@25 | PASS@100 |
| CL-7B + $\mathcal{D}_{original}$ | 6.5 | 9.5 | 15.0 |
| CL-7B + $\mathcal{D}_{modular}$ | 8.8 | 11.8 | 17.8 |
| CL-7B + $\mathcal{D}_{planning}$ | 6.9 | 10.5 | 15.4 |
| CL-7B + $\mathcal{D}_{plan}^{GT}$ | 17.9 | 22.3 | 28.1 |
| | +9.1 | +10.5 | +11.3 |

Table 5: **Effect of using ground-truth plans.** We disentangle the high-level reasoning vs coding capabilities by extracting ground-truth plans from solutions corresponding to the test problems. We find significant improvement in the performance on the CODE-CONTESTS-PLAN dataset, indicating that the model trained on the $\mathcal{D}_{planning}$ dataset while incapable of building correct plans, can follow such plans accurately.

**In-context Learning.** We first evaluate the performance of the model when provided with *parallel* two-shot in-context learning examples from $\mathcal{D}_{original}$ and $\mathcal{D}_{modular}$ datasets each. We find that the PASS@1 improves from 14.2 to 17.5 (a 23% relative improvement) on the APPS-INTRODUCTORY dataset and PASS@100 improves from 7.2 to 9.3 (a 29% relative improvement) on the CODE-CONTESTS dataset. These results indicate that more readablity and better-structured coding is helpful to the model in solving more problems.

**Fine-tuning.** Next, we fine-tune the model on the $\mathcal{D}_{original}$ and $\mathcal{D}_{modular}$ datasets and again find strong performance improvements from our transformation approach. Specifically, on the APPS-INTRODUCTORY dataset, the PASS@1 improves from 18.7 to 22.7 (a 23% relative improvement). Similarly, the CODE-CONTESTS dataset PASS@25 metric improves from 6.4 to 8.4 (30% relative improvement). These results cement our above findings about the effect of cleaning the data.

Interestingly, we also note that fine-tuning only provides modest improvements over the in-context learning performance. We hypothesize that this is due to the challenging nature of our task. [6]

### 4.2.2 EFFECT OF PLANNING ANNOTATIONS

Prior work has demonstrated considerable successes in improving reasoning in LLMs (Yue et al., 2023; Magister et al., 2022; Fu et al., 2023) by performing supervised learning on natural language *reasoning* or *planning steps*. We perform similar experiment, fine-tuning the model on $\mathcal{D}_{planning}$ dataset consisting of plans generated by our approach on top of $\mathcal{D}_{modular}$. We find that planning only provides a modest improvement over the $\mathcal{D}_{modular}$ dataset (PASS@25 improved from 42.6 to 43.9 on the APPS-INTRODUCTORY dataset) or often no improvements at all.

Upon inspection of the generated solutions, we find that often the generated plans are imprecise or incorrect, highlighting that planning still remains a bottleneck. To disentangle the high-level planning from the coding component, we analyze the performance of the model when provided with ground-truth plans on the CODE-CONTESTS dataset. We extract these ground-truth plans by applying our data transformation approach on the test set (similar to how $\mathcal{D}_{planning}$ training set was created). Table 5 provides results on this subset of 109 problems from the CODE-CONTESTS dataset for which we were able to extract the ground truth plans (since some problems don't have a valid `python` solutions). While our model trained on the $\mathcal{D}_{planning}$ dataset is incapable of synthesizing new plans, it can follow the generated plans correctly. All metrics improve significantly, e.g. PASS@100 improving from 17.8 to 28.1, well over the performance of GPT-3.5-TURBO, a much larger model!

---

[6]Note that the in-context examples add over 2,000 tokens to the prefix and lead to much slower decoding

```
def read_grid():
  n,m = input().split()
  ...

def remove_white_rows(grid):
  row_indices = []
  ...
  return grid

def remove_white_columns(grid):
  column_indices = []
  ...
  return grid

def main():
  grid = read_grid()
  grid = remove_white_rows(grid)
  grid = remove_white_columns(grid)
  print_grid(grid)
  ...
```

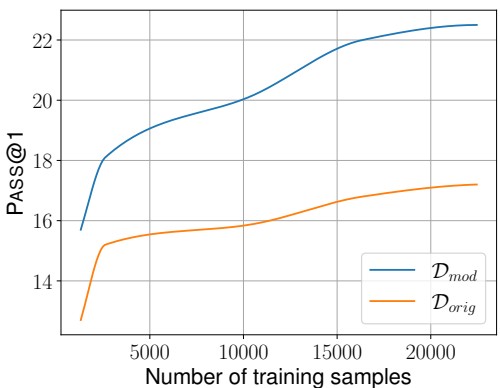

Figure 2: Example of a program generated by our model trained on the $\mathcal{D}_{modular}$ dataset. It solves the problem by using helper functions acting on rows and columns.

Figure 3: **Effect of quality on data-efficiency of the model**. Finetuning on 12.5% of clean $\mathcal{D}_{modular}$ dataset results in similar performance as finetuning on the entire $\mathcal{D}_{original}$ dataset.

Our mixed results raise critical questions for future work on improving planning in LLMs. In particular, poor performance might be attributed to any *imprecision* in automatically generated plans. Future data curation techniques that filter or augment this imprecision would be valueable. Alternatively, the supervised learning paradigm followed in this work might be insufficient for models to generalize *planning* in complex domains. Future work can explore alternative learning algorithms, possibly over our modularization approach which naturally decomposes programs.

### 4.2.3 ABLATIONS

**Effect of data size.** Beyond improving the quality of the resulting model, data quality is also attributed to improving the data efficiency. We evaluate this aspect by fine-tuning our model on different fractions of $\mathcal{D}_{original}$ and $\mathcal{D}_{modular}$ datasets and find similar results. Figure 3 presents the performance of the model as a function of training set size. As shown in the figure, training on just 15% of $\mathcal{D}_{modular}$ dataset achieves similar PASS@1 as fine-tuning on the entire $\mathcal{D}_{original}$.

**Effect of renaming.** We use variable renaming as an intermediate step in our cleaning process. We evaluate the performance of the model fine-tuned only on the $\mathcal{D}_{rename}$ dataset and find that renaming provides some performance improvements when compared to fine-tuning on $\mathcal{D}_{original}$ dataset. For example, PASS@1 improved from 17.2 to 19.1 on APPS-INTRODUCTORY. However, renaming still performs worse in comparison to fine-tuning on the $\mathcal{D}_{modular}$. This highlights that beyond just readable code, functional decomposition is also a key aspect of improving our performance.

**Cleaning Transformations vs Distillation.** We compare our transformation approach with a direct distillation baseline where we directly generate solutions using GPT-3.5-TURBO, referred to as the $\mathcal{D}_{distill}$ dataset[7]. This corresponds to various LLM instruction or fine-tuning approaches (Xu et al., 2023; Li et al., 2023b) providing a strong baseline for data cleaning. On the APPS-INTRODUCTORY dataset, we find that fine-tuning on the $\mathcal{D}_{modular}$ dataset achieves better performance compared to the $\mathcal{D}_{distill}$ dataset demonstrating the advantage of cleaning over the generation baseline.

**Choice of transformation model.** To evaluate how the choice of transformation model affects performance, we use the GPT-4-TURBO model to transform on a subset of the training set (detailed setup in Appendix C.3). GPT-4-TURBO, a stronger model, performs the transformations successfully and the resulting model trained on this version of the modularized dataset achieves even higher accuracy. For instance, PASS@10 improves from 33.0 when using $\mathcal{D}_{modular}$ constructed with GPT-3.5-TURBO to 34.3 when using the $\mathcal{D}_{modular}$ constructed with GPT-4-TURBO (full results in Table 8).

### 4.2.4 COMPARISON TO OTHER BASELINES

Beyond CL-7B, fine-tuned models outperform strong baselines like ALPHACODE on the CODE-CONTESTS dataset but still lag behind larger CODE-DAVINCI-002 and GPT-3.5-TURBO models.

---

[7]Note that we generate these solutions using in-context examples from the $\mathcal{D}_{modular}$ dataset

### 4.2.5 Case study of generated modularized program

Figure 2 provides an example of a program correctly generated by a model fine-tuned on our $\mathcal{D}_{modular}$ dataset. The problem requires removing rows and columns containing cells with certain attributes (i.e., if the cell is white) The modularized solution correctly identifies the steps required to solve the problem and implements them as separate helper functions, providing readable code.

## 5 Related Work

**Instruction tuning.** Instruction tuning refers to the process of finetuning a base pretrained LLM to perform general-purpose tasks and follow instructions. Recent works, Zhou et al. (2023); Cao et al. (2023); Chen et al. (2023a) have demonstrated that a small high-quality instruction corpus is sufficient for achieving good instruction tuning performance. Here, we perform task-specific fine-tuning of LLMs and observe similar performance improvements.

**Synthetic data for LLMs.** Recent works have explored using synthetic datasets for general-purpose or task-specific finetuning of LLMs. These approaches work by generating synthetic datasets from a strong LLM (like GPT-3.5-TURBO or GPT-4) using a set of existing tasks (Taori et al., 2023; Chiang et al., 2023) or generating new tasks using self-instruct (Wang et al., 2022) or evol-instruct (Xu et al., 2023) approaches. This has been also applied for task-specific finetuning – in common-sense reasoning (West et al., 2022), text-summarization (Sclar et al., 2022), mathematical reasoning (Luo et al., 2023a; Yue et al., 2023), tool use (Patil et al., 2023), coding (Luo et al., 2023b), and general-purpose reasoning Li et al. (2023b); Zelikman et al. (2022).

More specifically, Yue et al. (2023) curates diverse corpus of mathematics problems with chain-of-thought or program-of-thought (Chen et al., 2022b) annotations for mathematical reasoning analogous to our plans. Gunasekar et al. (2023) proposed pre-training models on programming "textbooks" generated synthetically from GPT-3.5-TURBO. Haluptzok et al. (2023) similarly generates programming puzzles and corresponding solutions from language models. Our work also studies curating synthetic data for code-generation space. However, instead of directly generating data using LLMs, we identify good programming patterns and clean existing datasets using them.

**Algorithmic Code Generation.** Code generation is a broad domain and is covered in Appendix D. We only discuss pertinent algorithmic code generation works here. Hendrycks et al. (2021) released the APPS dataset while Li et al. (2022) released the CODE-CONTESTS dataset with the ALPHACODE models. Zhang et al. (2023c) proposed a lookahead-search-based decoding algorithm for improving *reasoning* in LLMs and is orthogonal to our work. Chen et al. (2022a); Zhang et al. (2023b) proposed CODET and ALGO, that use generated tests to re-rank the generated solutions. Zelikman et al. (2023) proposed the PARSEL approach where they first generate a plan in a problem-specification language and then generate a program using it. Li et al. (2023a) also study disentangling the planning and coding for closed source LLMs, similar to our experiments on open models.

## 6 Discussion and Conclusion

Traditionally, data quality has been linked to functional correctness, ignoring the rich stylistic aspects differing across programs. In this work, we demonstrate that these aspects like readability, and program structuring actually impact the performance of the trained model on downstream tasks and thus also contribute to *data quality*, perhaps in relation to amenability to autoregressive modeling. Next, we proposed a novel data-cleaning pipeline demonstrating that LLMs can be used for transforming existing datasets to improve their quality based on user-instructions and oracle equivalence checker. While our evaluations focused on the algorithmic code generation task, we believe that this approach would also be useful for other domains for improving data quality as well. In particular, even in the absence of symbolic checkers (like test cases), we believe that there is an opportunity to use learned "oracles" for ensuring consistency and quality in other domains akin to how used in Sclar et al. (2022). Finally, beyond improving algorithmic code generation, we believe our modularization approach can be beneficial for general software engineering use cases (test generation, debugging, verification) where modularity is beneficial. A key limitation is that this work relies upon proprietary models for data transformation. With access to stronger open models, we believe our approach can also be applied in self-training setup using oracle equivalence checkers.

**Acknowledgement**    This work was supported in part by NSF grants CCF-1900968, CCF-1908870 and by SKY Lab industrial sponsors and affiliates Astronomer, Google, IBM, Intel, Lacework, Microsoft, Mohamed Bin Zayed University of Artificial Intelligence, Nexla, Samsung SDS, Uber, and VMware and finally generously provided research credits from Vast.ai. Any opinions, findings, conclusions, or recommendations in this paper are solely those of the authors and do not necessarily reflect the position of the sponsors. Additionally, we thank Alex Gu, Manish Shetty, and anonymous reviewers for helpful discussion and feedback on the paper.

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

## A  EXPERIMENTAL SETUP

**APPS benchmark.**  Since some of the problems in the APPS dataset are sourced from websites that provide insufficient or absent test cases, we filter the problems from those platforms. Specifically, we only retain problems from the `codeforces`, `codechef`, and `atcoder` competition websites. This also removes the disparity/domain-shift between the training and test splits which has been observed as an issue in the APPS dataset in prior works (Section 4.1 in Li et al. (2023c)). While we considerably reduced the size of our training set, our test set is still quite close to the test set containing around 3800 problems instead of the default 5000.

**CODE-CONTESTS benchmark.**  The original CODE-CONTESTS benchmark consists of 13,328 problems in the training dataset. We restrict the dataset to only problems with valid `python` solutions that pass the test cases. Next, since the original dataset provides over 100 solutions per problem, we perform minhash-based deduplication on the solutions (hash size=64, num bands=60, band size=5) from gaoya[8] and retain a maximum of 25 solutions per problem. This results in about 7k problems in the training set spanning about 98.5k solutions. We do not perform any filtering on the test set.

Additionally, we note that some of the provided solutions in both APPS and CODE-CONTESTS datasets do not pass the test cases. These cases are sometimes caused by incorrect programs, often because of scraping solutions with the wrong language. However, more often problems in these datasets support multiple correct solutions (for instance solutions to a problem can return a list of elements in any order). The ground truth test cases only check for a single correct solution and thus result in many solutions failing the test cases. We retain such samples for the smaller APPS dataset and instead check whether the transformed program behavior is similar to the original program.

**Metrics**  We use the PASS@$K$ to perform our evaluations. We perform nucleus sampling using vLLM with $p = 0.95$. We outline the default sampling configurations used for computing the metrics

1. PASS@1 - We use a sampling budget ($N$) = 10 and temperature = 0.1.
2. PASS@10 - We use a sampling budget ($N$) = 50 and temperature = 0.6.
3. PASS@25 - We use a sampling budget ($N$) = 50 and temperature = 0.6.
4. PASS@100 - We use a sampling budget ($N$) = 200 and temperature = 0.8.

**Finetuning details**  We finetune the CODELLAMA-7B model using deepspeed huggingface trainer. We use the following training configuration for our main experiments -

| Training Parameters | Values |
|---|---:|
| LR | $5e^{-5}$ |
| Epochs | 1 or 2 depending on the dataset |
| Batch Size | 256 (combing grad. accumulation) |
| Dtype | bf16 |

---

[8]`https://github.com/serega/gaoya`

# B  CODE TRANSFORMATIONS IMPLEMENTATION

We implement our code transformation approach using zero-shot prompting with GPT-3.5-TURBO model. After transformation, we extract the generated code and evaluate its functional correctness using the provided test cases. In case the program does not pass, we retry the process with up to a maximum of 5 attempts. In our experience, instruction-tuned models can follow precise commands and transform programs very well.

## B.1  RENAMING

We use the following prompt to perform renaming.

```
QUESTION:
{problem_statement}

ANSWER:
```python
{solution}
```
Rename the variables in the program to be descriptive, meaningful, and consistent. Do not
change the original semantics of the program. Enclose the program within backticks as shown
above and remember to use descriptive variable names.
```

## B.2  MODULARIZATION

Unlike renaming, we perform two rounds of modularization in case the generated program consists of long function implementations (hinting that the function can be decomposed further). We use the following prompt to perform the first round of modularization

```
QUESTION:
{problem_statement}

ANSWER:
```python
{renamed_solution}
```
Refactor the above program. Follow the guidelines
* make the program more modular with smaller and meaningful helper functions
* good descriptive names for the helper functions
* have an entry function called `main()`
* `main()` is called inside `if __name__ == '__main__'`

Do not change the original semantics of the program significantly and no need to perform
optimizations. Enclose the program within backticks as shown above
```

Next, in case the modularized program contains a function with the number of lines greater than 20, we further prompt the model while signaling which functions to further decompose. This occurs in about 20-40% of modularized solutions and we use the following prompt.

```
QUESTION:
{problem_statement}

ANSWER:
```python
{modularized_solution}
```
Refactor the above program by modularizing it and breaking down long and complex functions
into smaller meaningful helper functions. Particularly refactor and decompose the following
function(s) into smaller helper functions – {function_names_string}
Only return the refactored program enclosed in backticks as shown above."""
```

## B.3  PLANNING

We use the following prompt to generate natural language plans

```
QUESTION:
{problem_statement}

ANSWER:
```python
{modularized_solution}
```
Generate a summary for the following functions and classes in the program within four lines
each. The summaries should be descriptive and helpful for understanding the program (however
yet concise in four lines).
The functions and classes are –
{list_of_function_names}
Follow the provided format for the summaries while being informative and concise. Enclose the
signatures in backticks as shown above.
```

# C ADDITIONAL RESULTS

## C.1 GPT-4 JUDGE EVALUATION FOR THE TRANSFORMATIONS

We here present some quantitative evidence of the improvements made from our transformation approach. However, since the transformations are free-form code generation, we rely on using GPT-4 *as judge*, an evaluation approach gaining popularity for evaluating free-form language outputs (Zheng et al., 2023; Zhuo, 2023). Specifically, we ask the language model to answer whether the modularized refactored code has better variable names, better function decomposition, and is consistent with the original program. The model can provide answers on a key of 1-3 from comparison questions and 0-1 for the consistency question. The following prompt depicts our approach

```
SYSTEM PROMPT:
Please act as an impartial judge and evaluate the code refactoring below. You need to
evaluate whether the refactored program uses better and correct variable names, refactors the
 implementation into correct smaller helper functions and consistency with the original
program. Your evaluation should be based on correctnes and helpfulness of the refactoring in
better understanding the problem and also if it is still consistent with the original program
, i.e. it follows similar program logic and algorithm.

* For evaluating variable names and function decomposition, please give a score from 1 to 3
where 1 means the refactoring is not helpful at all, 2 means the refactoring is somewhat
helpful and 3 means the refactoring is very helpful. Example format

Variable names reasoning: [[reasoning for the variable names score, often assessing whether
the variable names are more descriptive and meaningful and correctly reflect the variable's
purpose]]
Variable names: [[1]] or [[2]] or [[3]]

Function decomposition reasoning: [[reasoning for the decomposition score, often assessing
whether some function is too long, possibility to perform further abstractions, choice of
abstractions, helper function names]]
Function decomposition: [[1]] or [[2]] or [[3]]

* For evaluating consistency, please give a score of 0 if the refactored program is not
consistent with the original program and 1 if it is consistent. Example format

Consistency reasoning: [[reasoning for the consistency score, often assessing whether the
refactored program follows similar program logic and algorithm as the original program]]
Consistency: [[0]] or [[1]]

QUESTION:
{problem_statement}

ORIGINAL SOLUTION:
{solution}

REFACTORED SOLUTION:
{solution}
```

While this evaluation might portray certain subtle biases, we believe it still provides us a signal to assess the quality of the transformations. To reduce costs, we the GPT-4 as judge evaluation to 1000 problems in the APPS dataset [9]. GPT-4 followed the proposed format for 998 solutions and we present results on them in Table 6. Results demonstrate that most of the applied transformations are meaningful while remaining consistent with the original ground truth solutions.

|  | Score distribution | Average |
|---|---|---|
| Variable names | $\{3 : 967, 2 : 28, 1 : 3\}$ | 2.96 |
| Function decomposition | $\{3 : 938, 2 : 59, 1 : 1\}$ | 2.93 |
| Consistency | $\{1 : 994, 0 : 4\}$ | 0.994 |

Table 6: **GPT-4 as a judge evaluation for quality of 998 transformed examples**. We compared the original and unmodified solution using GPT-4 for variable names used, function decomposition, and consistency of the modified and original solution. Results demonstrate that most of the transformations are successful and meaningful while being consistent with the original program.

---

[9]Our prompt spans around 1.5-2k tokens including problem, original, and refactored programs leading to high costs

To get better insights into GPT-4 evaluations, we look at the examples which receive lower scores. The scores and associated reasoning appear meaningful. For example, in Figure 14, the modularized program is already significantly more readable than original renamed program. However, GPT-4 identifies that the `calculate_permutation_even` and `calculate_permutation_even` helper functions are virtually the same and can be abstracted further. Note that this transformation is an artifact of the fact that original program consisted of same program logic distributed across two far apart if conditions. Similarly, in Figure 15, GPT-4 identified some unmodified variable names like `t` while acknowledging other improvements such as `sky` to `heights` giving it a rating of 2. The rating of 1 is provided when the transformation does not modify any variable names or does not decompose existing functions, as evidenced by score distributed, a rare occurrence. Indeed, often the examples marked with a rating 2 actually improve upon orignal code in non-trivial ways.[10]

## C.2 APPS-COMPETITION RESULTS

We present the results on APPS-COMPETITION dataset here.

| | APPS-COMPETITION | | |
|---|---|---|---|
| | PASS@1 | PASS@10 | PASS@100 |
| **Fine-tuning** | | | |
| CL-7B + $\mathcal{D}_{original}$ | 0.2 | 1.7 | 3.1 |
| CL-7B + $\mathcal{D}_{modular}$ | 0.5 | 2.3 | 3.2 |
| | **+0.3** | **+0.6** | **+0.1** |
| CODE-DAVINCI-002 | 0.3 | 2.9 | 5.7 |

Table 7: **Results on the APPS-COMPETITION dataset.**

## C.3 ABLATION ON CHOICE OF MODEL

We use GPT-3.5-TURBO as the default model for performing the transformations in the main experiments since it provides a nice balance between the accuracy and cost of performing the transformations. Here, to demonstrate the generality of our approach we perform an ablation by replacing the transformation model with GPT-4-TURBO. Since this model is about 8-10x more expensive than GPT-3.5-TURBO, we perform this ablation on a subset of 5k programs sampled from the dataset.

**Experimental setup.** We repeat the renaming and modularization steps described in Section 3.2 using the GPT-4-TURBO model. We call the resulting transformed dataset as $\mathcal{D}^4_{modular}$. Next, fairly compare the resulting dataset with the original and modularized dataset generated using GPT-3.5-TURBO, we sample the corresponding parallel original and transformed programs and call them $\mathcal{D}_{original}$ and $\mathcal{D}^{3.5}_{modular}$ datasets.

| | APPS-INTRODUCTORY | | |
|---|---|---|---|
| | PASS@1 | PASS@10 | PASS@100 |
| CL-7B + $\mathcal{D}_{original}$ | 16.3 | 31.6 | 37.6 |
| CL-7B + $\mathcal{D}^{3.5}_{modular}$ | 18.8 | 33.0 | 38.2 |
| CL-7B + $\mathcal{D}^4_{modular}$ | 19.4 | 34.3 | 40.0 |
| | **+0.6** | **+1.3** | **+1.8** |

Table 8: **Ablation on the choice of model used for performing the transformations.** $\mathcal{D}^{3.5}_{modular}$ represents the dataset generated using GPT-3.5-TURBO and $\mathcal{D}^4_{modular}$ represents the dataset generated using GPT-4-TURBO. We find that the performance of the model trained on the $\mathcal{D}^4_{modular}$ dataset is better than the model trained on the $\mathcal{D}^{3.5}_{modular}$ dataset.

---

[10]Curiously, GPT-4 sometimes returned a rating of 2.5 instead of integer 2 or 3. We rounded it to 2, thus making our evaluation harsher!

## D  ADDITIONAL RELATED WORK

Code LLMs have been used for multiple domains in various lines of approaches. Here, we present a few key approaches and recommend the reader to Hou et al. (2023) for a detailed survey. Chen et al. (2021) released the CODE-DAVINCI-002 model and evaluate it for code generation. Since then, LLMs have been used for a variety of domains such as data science (Jain et al.; Lai et al., 2022; Yin et al., 2023), APIs (Zhiruo Wang & Neubig, 2022; Patil et al., 2023), and repositories (Zhang et al., 2023a; Bairi et al., 2023; Shrivastava et al., 2023). (Le et al., 2022; Shojaee et al., 2023; Liu et al., 2023b) use reinforcement learning with compilation/execution feedback to fine-tune code LLMs for (algorithmic) code generation task.

Other works have approached code generation from different fronts, exploring planning (Jiang et al., 2023), repair (Chen et al., 2023b; Shinn et al., 2023), test generation (Key et al., 2022; Chen et al., 2022a), and prompt optimization (Liu et al., 2023a). . Le et al. (2023) proposed a prompting-based approach for modular code-generation.

# E  EXAMPLES OF TRANSFORMED PROGRAM

```python
def main():
    import sys
    input=sys.stdin.readline

    n,k=map(int,input().split())

    ab=[list(map(int,input().split())) for _ in
      [0]*n]

    g=[[] for _ in [0]*10]
    [g[b-1].append(a) for a,b in ab]
    [g[c].sort(reverse=True) for c in range(10)]

    for c in range(10):
        g[c]=[0]+g[c]

    for c in range(10):
        for i in range(2,len(g[c])):
            g[c][i]+=g[c][i-1]+2*(i-1)

            dp=[0]*(k+1)

    for c in range(10):
        dp2=[0]*(k+1)
        for i in range(len(g[c])):
            for j in range(k+1-i):
                dp2[i+j]=max(dp2[i+j],dp[j]+g[c][i
    ])
        dp=dp2

        print(max(dp))

        if __name__=='__main__':
    main()
```

(a) Original program

```python
def read_input():
    ... (TRUNCATED)
    return num_books, num_sell, book_info

def group_books_by_genre(num_books, book_info):
    genre_books = [[] for _ in range(10)]
    for price, genre in book_info:
        genre_books[genre-1].append(price)
    return genre_books

def sort_books_by_price(genre_books):
    for genre in genre_books:
        genre.sort(reverse=True)
    return genre_books

def calculate_purchase_prices(genre_books):
    for genre in genre_books:
        genre.insert(0, 0)
    for genre in genre_books:
        for i in range(2, len(genre)):
            genre[i] += genre[i-1] + 2*(i-1)
    return genre_books

def calculate_max_purchase_price(num_sell,
    genre_books):
    dp = [0] * (num_sell+1)
    for genre in genre_books:
        dp2 = [0] * (num_sell+1)
        for i in range(len(genre)):
            for j in range(num_sell+1-i):
                dp2[i+j] = max(dp2[i+j], dp[j] +
    genre[i])
        dp = dp2
    return max(dp)

def main():
    num_books, num_sell, book_info = read_input()
    genre_books = group_books_by_genre(num_books,
      book_info)
    genre_books = sort_books_by_price(genre_books)
    genre_books = calculate_purchase_prices(
      genre_books)
    max_purchase_price =
      calculate_max_purchase_price(num_sell,
      genre_books)
    print(max_purchase_price)

if __name__ == '__main__':
    main()
```

(b) Transformed program

Figure 4: Original and transformed programs

```python
for _ in range(int(input())):
    p, q = map(int,input().split())
    c = q
    d = p

i = 1
factor = []
while i*i <= q:
    if q % i == 0:
        factor.append(i)
        if q//i != i:
            factor.append(q//i)
    i += 1

factor.sort(reverse=True)
factor.pop()
m = 1
for i in factor:
    d = p
    while d % c == 0:
        d //= i
    m = max(m, d)
print(m)
```

```python
def get_factors(q):
    factors = []
    i = 1
    while i*i <= q:
        if q % i == 0:
            factors.append(i)
            if q//i != i:
                factors.append(q//i)
        i += 1
    factors.sort(reverse=True)
    factors.pop()
    return factors

def get_largest_divisor(p, q, factors):
    largest_divisor = 1
    for i in factors:
        dividend_copy = p
        while dividend_copy % q == 0:
            dividend_copy //= i
        largest_divisor = max(largest_divisor,
    dividend_copy)
    return largest_divisor

def main():
    t = int(input())
    for _ in range(t):
        p, q = map(int, input().split())
        factors = get_factors(q)
        largest_divisor = get_largest_divisor(p, q,
    factors)
        print(largest_divisor)

if __name__ == '__main__':
    main()
```

(a) Original program          (b) Transformed program

Figure 5: Original and transformed programs

```python
def ncr(n, r, p):
    # calculate nCr modulo p
    # initialize numerator and denominator
    numerator = denominator = 1
    for i in range(r):
        numerator = (numerator * (n - i)) % p
        denominator = (denominator * (i + 1)) % p
    return (numerator * pow(denominator, p - 2, p))
        % p

def is_good_number(number, a, b):
    # check if a number is a good number
    while number != 0:
        if number % 10 != a and number % 10 != b:
            return False
        number //= 10
    return True

def count_excellent_numbers(a, b, n):
    ans = 0
    p = 10**9 + 7
    numerator = 1
    denominator = 1

    for i in range(n + 1):
        sum_of_digits = a * i + b * (n - i)

        if i != 0:
            numerator = (numerator * (n - i + 1)) %
    p
            denominator = (denominator * i) % p

        if is_good_number(sum_of_digits, a, b):
            ans = (ans + (numerator * pow(
    denominator, p - 2, p)) % p) % p

    return ans % p

def main():
    a, b, n = map(int, input().split())
    result = count_excellent_numbers(a, b, n)
    print(result)

if __name__ == '__main__':
    main()
```

```python
def ncr(n, r, p):
    # initialize numerator
    # and denominator
    num = den = 1
    for i in range(r):
        num = (num * (n - i)) % p
        den = (den * (i + 1)) % p
    return (num * pow(den,
            p - 2, p)) % p
a,b,n=map(int,input().split())
ans=0
p=10**9+7
num=1
den=1
for i in range(n+1):
    s=a*i+b*(n-i)
    if i!=0:
        num=(num*(n-i+1))%p
        den=(den*(i))%p
    am=True
    while s!=0:
        if s%10!=a and s%10!=b:
            am=False
            break
        s//=10
    if am:
        ans=(ans+(num*pow(den,p-2,p))%p)%p
print(ans%p)
```

(a) Original program                (b) Transformed program

Figure 6: Original and transformed programs

```
import bisect
rev=[]
for i in range(1,10002):
    if str(i)==str(i)[::-1]:rev.append(i)
n=int(input())
ind= bisect.bisect_left(rev,n)
if abs(n-rev[ind-1])<=abs(n-rev[ind]):
    print(rev[ind-1])
else:
    print(rev[ind])
```

(a) Original program

```
import bisect

def generate_palindromes():
    palindromes = []
    for num in range(1, 10002):
        if str(num) == str(num)[::-1]:
            palindromes.append(num)
    return palindromes

def find_closest_palindrome(palindromes, n):
    index = bisect.bisect_left(palindromes, n)
    if abs(n - palindromes[index - 1]) <= abs(n -
     palindromes[index]):
        return palindromes[index - 1]
    else:
        return palindromes[index]

def main():
    palindromes = generate_palindromes()
    n = int(input())
    closest_palindrome = find_closest_palindrome(
     palindromes, n)
    print(closest_palindrome)

if __name__ == '__main__':
    main()
```

(b) Transformed program

Figure 7: Original and transformed programs

```python
def find_divisors(max_interval):
    divisors = []
    for divisor in range(1, max_interval+1):
        if max_interval % divisor == 0:
            divisors.append(divisor)
    return divisors

def calculate_min_sum(intervals, divisors):
    current_index = min_sum = 0
    for interval in intervals:
        while divisors[current_index] < interval:
            current_index += 1
        min_sum += divisors[current_index] -
     interval
    return min_sum

def main():
    num_metronomes = int(input())
    intervals = [int(input()) for i in range(
     num_metronomes)]
    intervals.sort()
    if num_metronomes == 1:
        min_sum = 0
    else:
        max_interval = intervals[-1]
        divisors = find_divisors(max_interval)
        min_sum = calculate_min_sum(intervals,
     divisors)
    print(min_sum)

if __name__ == '__main__':
    main()
```

```python
N = int(input())
T = [int(input()) for i in range(N)]
T.sort()
if N == 1:
    ans = 0
else:
    y = T[-1]
    R = []
    for x in range(1, y+1):
        if y % x == 0:
            R.append(x)
    cur = ans = 0
    for t in T:
        while R[cur] < t:
            cur += 1
        ans += R[cur] - t
print(ans)
```

(a) Original program          (b) Transformed program

Figure 8: Original and transformed programs

```python
def main():
    from collections import defaultdict
    n, colors = int(input()), input()[::2]
    dsu, edges, d = list(range(n)), [], defaultdict(list)
    for _ in range(n - 1):
        u, v = map(int, input().split())
        u -= 1
        v -= 1
        if colors[u] == colors[v]:
            a, b = dsu[u], dsu[v]
            while a != dsu[a]:
                a = dsu[a]
            while b != dsu[b]:
                b = dsu[b]
            if a < b:
                dsu[b] = dsu[v] = a
            else:
                dsu[a] = dsu[u] = b
        else:
            edges.append(u)
            edges.append(v)
    for u, v in enumerate(dsu):
        dsu[u] = dsu[v]
    while edges:
        u, v = dsu[edges.pop()], dsu[edges.pop()]
        d[u].append(v)
        d[v].append(u)

    def bfs(x):
        nxt, avail, t = [x], [True] * n, 0
        while nxt:
            t += 1
            cur, nxt = nxt, []
            for y in cur:
                avail[y] = False
                for y in d[y]:
                    if avail[y]:
                        nxt.append(y)
        return t if x else cur[0]

    print(bfs(bfs(0)) // 2)

if __name__ == '__main__':
    main()
```

(a) Original program

```python
from collections import defaultdict

def find_root(vertex, dsu):
    while vertex != dsu[vertex]:
        vertex = dsu[vertex]
    return vertex

def merge_trees(u, v, dsu):
    root_u = find_root(u, dsu)
    root_v = find_root(v, dsu)
    if root_u < root_v:
        dsu[root_v] = dsu[v] = root_u
    else:
        dsu[root_u] = dsu[u] = root_v

def build_graph(num_vertices, colors, edges):
    dsu = list(range(num_vertices))
    graph = defaultdict(list)
    for u, v in edges:
        if colors[u] == colors[v]:
            merge_trees(u, v, dsu)
        else:
            graph[dsu[u]].append(dsu[v])
            graph[dsu[v]].append(dsu[u])
    return dsu, graph

def bfs(x, num_vertices, graph):
    next_vertices = [x]
    available = [True] * num_vertices
    t = 0
    while next_vertices:
        t += 1
        current_vertices, next_vertices = next_vertices, []
        for y in current_vertices:
            available[y] = False
            for neighbor in graph[y]:
                if available[neighbor]:
                    next_vertices.append(neighbor)
    return t if x else current_vertices[0]

def main():
    num_vertices = int(input())
    colors = input()[::2]
    edges = []
    for _ in range(num_vertices - 1):
        u, v = map(int, input().split())
        u -= 1
        v -= 1
        edges.append((u, v))
    dsu, graph = build_graph(num_vertices, colors, edges)
    print(bfs(bfs(0, num_vertices, graph),
        num_vertices, graph) // 2)

if __name__ == '__main__':
    main()
```

(b) Transformed program

Figure 9: Original and transformed programs

```python
import heapq

def dfs(graph, start):
    n = len(graph)
    dist = [-0 for i in range(n + 1)]
    visited = [False for i in range(n + 1)]
    visited[start] = True
    stack = []
    dist[start] = 0
    heapq.heappush(stack, start)
    while stack:
        u = heapq.heappop(stack)
        for v in graph[u]:
            if not visited[v]:
                visited[v] = True
                dist[v] = dist[u] + 1
                heapq.heappush(stack, v)
    return dist

def solution():
    n, m, d = map(int, input().strip().split())
    p = list(map(int, input().strip().split()))
    graph = [[] for i in range(n + 1)]
    for i in range(n - 1):
        a, b = map(int, input().strip().split())
        graph[a].append(b)
        graph[b].append(a)
    dist = dfs(graph, 1)

    max_distance = -1
    u = -1
    v = -1
    for i in p:
        if dist[i] > max_distance:
            max_distance = dist[i]
            u = i

    distu = dfs(graph, u)

    max_distance = -1
    for i in p:
        if distu[i] > max_distance:
            max_distance = distu[i]
            v = i

    distv = dfs(graph, v)

    affected = 0
    for i in range(1, n + 1):
        if 0 <= distu[i] <= d and 0 <= distv[i] <=
     d:
            affected += 1

    print(affected)

solution()
```

(a) Original program

```python
import heapq

def calculate_distances(graph, start):
    n = len(graph)
    distances = [-0 for i in range(n + 1)]
    visited = [False for i in range(n + 1)]
    visited[start] = True
    stack = []
    distances[start] = 0
    heapq.heappush(stack, start)
    while stack:
        current_node = heapq.heappop(stack)
        for neighbor in graph[current_node]:
            if not visited[neighbor]:
                visited[neighbor] = True
                distances[neighbor] = distances[
     current_node] + 1
                heapq.heappush(stack, neighbor)
    return distances

def find_possible_book_locations():
    n, m, d = map(int, input().strip().split())
    affected_settlements = list(map(int, input().
      strip().split()))
    graph = [[] for i in range(n + 1)]
    for i in range(n - 1):
        a, b = map(int, input().strip().split())
        graph[a].append(b)
        graph[b].append(a)
    return calculate_possible_locations(n, m, d,
      affected_settlements, graph)

def calculate_possible_locations(n, m, d,
      affected_settlements, graph):
    distances = calculate_distances(graph, 1)
    max_distance, u = find_max_distance(distances,
      affected_settlements)
    distances_u = calculate_distances(graph, u)
    max_distance, v = find_max_distance(distances_u
      , affected_settlements)
    distances_v = calculate_distances(graph, v)
    return count_possible_locations(n, d,
      distances_u, distances_v)

def find_max_distance(distances,
      affected_settlements):
    max_distance = -1
    u = -1
    for settlement in affected_settlements:
        if distances[settlement] > max_distance:
            max_distance = distances[settlement]
            u = settlement
    return max_distance, u

def count_possible_locations(n, d, distances_u,
      distances_v):
    possible_locations = 0
    for i in range(1, n + 1):
        if 0 <= distances_u[i] <= d and 0 <=
      distances_v[i] <= d:
            possible_locations += 1
    return possible_locations

def main():
    possible_locations =
      find_possible_book_locations()
    print(possible_locations)

if __name__ == '__main__':
    main()
```

(b) Transformed program

Figure 10: Original and transformed programs

```python
n = int(input())
dividers = list()
i = 2

while i * i <= n:
    if n % i == 0:
        dividers.append(i)

        while n % i == 0:
            n /= i

    i += 1

if n > 1:
    dividers.append(n)

if len(dividers) == 1:
    print(dividers[0])
else:
    print(1)
```

(a) Original program

```python
def find_dividers(path_length):
    dividers = []
    divisor = 2

    while divisor * divisor <= path_length:
        if path_length % divisor == 0:
            dividers.append(divisor)
            while path_length % divisor == 0:
                path_length //= divisor
        divisor += 1

    if path_length > 1:
        dividers.append(path_length)

    return dividers

def calculate_max_colors(path_length):
    dividers = find_dividers(path_length)
    if len(dividers) == 1:
        return dividers[0]
    else:
        return 1

def main():
    path_length = int(input())
    max_colors = calculate_max_colors(path_length)
    print(max_colors)

if __name__ == '__main__':
    main()
```

(b) Transformed program

Figure 11: Original and transformed programs

```
import sys
readline = sys.stdin.readline

N, M = map(int, readline().split())
mod = 10**9+7
dpscc = [[0]*(N+1) for _ in range(N+1)]
dpus = [[0]*(N+1) for _ in range(N+1)]
dpscc[1][0] = 1

for m in range(M):
    dpscc2 = [[0]*(N+1) for _ in range(N+1)]
    dpus2 = [[0]*(N+1) for _ in range(N+1)]
    for i in range(1, N+1):
        for j in range(N+1-i):
            kscc = dpscc[i][j]
            kus = dpus[i][j]
            dpscc2[i][j] = (dpscc2[i][j] + i*kscc)
    % mod
            dpus2[i][j] = (dpus2[i][j] + j*(kus+
    kscc)) % mod
            dpscc2[i+j][0] = (dpscc2[i+j][0] + i*
    kus) % mod
            if N-i-j:
                dpus2[i][j+1] = (dpus2[i][j+1] + (N
    -i-j)*(kus+kscc)) % mod

    dpscc = [d[:] for d in dpscc2]
    dpus = [d[:] for d in dpus2]
print(dpscc[N][0])
```

(a) Original program

```
import sys

def count_sequences(num_towns, num_days):
    mod = 10**9+7
    dp_same_city_count = [[0]*(num_towns+1) for _
        in range(num_towns+1)]
    dp_unique_city_count = [[0]*(num_towns+1) for _
        in range(num_towns+1)]
    dp_same_city_count[1][0] = 1

    for day in range(num_days):
        dp_same_city_count2 = [[0]*(num_towns+1)
        for _ in range(num_towns+1)]
        dp_unique_city_count2 = [[0]*(num_towns+1)
        for _ in range(num_towns+1)]
        for i in range(1, num_towns+1):
            for j in range(num_towns+1-i):
                same_city_count =
        dp_same_city_count[i][j]
                unique_city_count =
        dp_unique_city_count[i][j]
                dp_same_city_count2[i][j] = (
        dp_same_city_count2[i][j] + i*same_city_count
        ) % mod
                dp_unique_city_count2[i][j] = (
        dp_unique_city_count2[i][j] + j*(
        unique_city_count+same_city_count)) % mod
                dp_same_city_count2[i+j][0] = (
        dp_same_city_count2[i+j][0] + i*
        unique_city_count) % mod
                if num_towns-i-j:
                    dp_unique_city_count2[i][j+1] =
        (dp_unique_city_count2[i][j+1] + (num_towns-
        i-j)*(unique_city_count+same_city_count)) %
        mod

        dp_same_city_count = [d[:] for d in
        dp_same_city_count2]
        dp_unique_city_count = [d[:] for d in
        dp_unique_city_count2]

    return dp_same_city_count[num_towns][0]

def main():
    num_towns, num_days = map(int, input().split())
    result = count_sequences(num_towns, num_days)
    print(result)

if __name__ == '__main__':
    main()
```

(b) Transformed program

Figure 12: Original and transformed programs

```
def calculate_beauty(num_points, num_segments,
    segments):
    tail_length = [0]*(num_points+1)
    tail_length[1] = 1
    for i in range(2, num_points+1):
        temp = [tail_length[j] for j in segments[i
    ]]+[0]
        tail_length[i] = max(temp)+1
    spine_length = [len(segments[i])*tail_length[i]
        for i in range(1, num_points+1)]
    return max(spine_length)

def main():
    num_points, num_segments = [int(i) for i in
      input().split()]
    segments = {i:[] for i in range(1, num_points
      +1)}
    for j in range(num_segments):
        point1, point2 = [int(i) for i in input().
      split()]
        segments[point1].append(point2)
        segments[point2].append(point1)
    result = calculate_beauty(num_points,
      num_segments, segments)
    print(result)

if __name__ == '__main__':
    main()
```

```
n,m = [int(i) for i in input().split()]
seg = {i:[] for i in range(1,n+1)}
for j in range(m):
    a,b = [int(i) for i in input().split()]
    seg[a].append(b)
    seg[b].append(a)
tail = [0]*(n+1)
tail[1] = 1
for i in range(2,n+1):
    temp = [tail[j] for j in seg[i]]+[0]
    tail[i] = max(temp)+1
temp = [len(seg[i])*tail[i] for i in range(1,n+1)]
print(max(temp))
```

(a) Original program          (b) Transformed program

Figure 13: Original and transformed programs

```
def find_permutation
(n, k):
    if n % 2 == 0:
        if k < n * (n + 1) // 2 - 1 or k > 3 * (n
       // 2) ** 2 - 1:
            return None
        elif k == n * (n + 1) // 2 - 1:
            return list(range(1, n + 1))
        else:
            permutation =
        calculate_permutation_even(n, k)
            return permutation
    elif n == 1:
        return [1] if k == 0 else None
    elif k < n * (n + 1) // 2 - 1 or k > 3 * (n //
      2) * (n // 2 + 1):
        return None
    elif k == n * (n + 1) // 2 - 1:
        return list(range(1, n + 1))
    else:
        permutation = calculate_permutation_odd(n,
         k)
        return permutation

def calculate_permutation_even(n, k):
    k, count, p, l, x = k - n * (n + 1) // 2 + 1,
      0, 0, [0 for i in range(n)], 1
    while k > 0:
        p += 2
        k, count = k - n + p, count + 1
    for i in range(n, n - count, -1):
        l[x] = i
        x += 2
    k = -k
    l[2 * count - 1 + k], p = n - count + 1, 1
    for i in range(n):
        if l[i] == 0:
            l[i] = p
            p += 1
    return l

def calculate_permutation_odd(n, k):
    k, count, p, l, x = k - n * (n + 1) // 2 + 1,
      0, 0, [0 for i in range(n)], 1
    while k > 0:
        p += 2
        k, count = k - n + p, count + 1
    for i in range(n, n - count, -1):
        l[x] = i
        x += 2
    k = -k
    l[2 * count - 1 + k], p = n - count + 1, 1
    for i in range(n):
        if l[i] == 0:
            l[i] = p
            p += 1
    return l

def main():
    t = int(input())
    for _ in range(t):
        n, k = map(int, input().split())
        permutation = find_permutation(n, k)
        if permutation is not None:
            print(*permutation)
        else:
            print(-1)

main()
```

```
for i in range(int(input())):
    n,k=[int(i) for i in input().split()]
    if(n%2==0):
     if(k<(n*(n+1))//2 - 1  or  k>3*((n//2)**2) -
      1):
        print(-1)
     elif(k==(n*(n+1))//2 - 1):
      for i in range(1,n+1):
        print(i,'',end='')
      print()
     else:
      k,count,p,l,x = k-(n*(n+1))//2 + 1,0,0,[0 for
       i in range(n)],1
      while(k>0):
        p+=2 ;k,
        count = k-n+p ,count+1
      for i in range(n,n-count+1,-1):
        l[x]=i ;x+=2
      k=-k ;
      l[2*count - 1 +k],p = n-count+1 ,1
      for i in range(n):
       if(l[i]==0):
         l[i]=p
         p+=1
      for i in l:
        print(i,'',end='')
      print()
    else:
     if(n==1):
       print(1) if(k==0) else print(-1)
     elif(k<(n*(n+1))//2 - 1  or k>3*(n//2)*(n//2 +
       1)):
        print(-1)
     elif(k==(n*(n+1))//2 - 1):
      for i in range(1,n+1):
        print(i,'',end='')
      print()
     else:
      k,count,p,l,x = k-(n*(n+1))//2 + 1,0,0,[0 for
       i in range(n)],1
      while(k>0):
        p+=2 ;  k,count = k-n+p ,count+1
```

|  (a) Original program  |  (b) Transformed program  |

Figure 14: Original and transformed programs

```
import sys

num=int(sys.stdin.readline())
s=sys.stdin.readline().split()
sky=list(map(int,s))
sky.reverse()
cuts=0
change=0
t=False
i=1

while i<len(sky):
 if sky[i]<=sky[i-1]:
  for j in range(i-1,-1,-1):

   if sky[j]<=sky[i]-(i-j):
    break
   else:
    change+=sky[j]-(sky[i]-(i-j))

   if change>=sky[i]:
    change=sky[i]
    t=True
    break

  cuts+=change

  if t:
   del sky[i]
   t=False
   i-=1

  else:
   for j in range(i-1,-1,-1):
    if sky[j]<sky[i]-(i-j):
     break
    else:
     sky[j]=sky[i]-(i-j)
 i+=1

 change=0

print(cuts)
```

(a) Original program

```
import sys

def count_operations(heights):
    cuts = 0
    change = 0
    t = False
    i = 1
    while i < len(heights):
        if heights[i] <= heights[i - 1]:
            change = calculate_change(heights, i)
            cuts += change
            if t:
                del heights[i]
                t = False
                i -= 1
            else:
                update_heights(heights, i)
        i += 1
        change = 0
    return cuts

def calculate_change(heights, i):
    change = 0
    t = False
    for j in range(i - 1, -1, -1):
        if heights[j] <= heights[i] - (i - j):
            break
        else:
            change += heights[j] - (heights[i] - (i
    - j))
        if change >= heights[i]:
            change = heights[i]
            t = True
            break
    return change

def update_heights(heights, i):
    for j in range(i - 1, -1, -1):
        if heights[j] < heights[i] - (i - j):
            break
        else:
            heights[j] = heights[i] - (i - j)

def main():
    num_sky_scrappers = int(sys.stdin.readline())
    heights = list(map(int, sys.stdin.readline().
      split()))
    heights.reverse()
    cuts = count_operations(heights)
    print(cuts)

main()
```

(b) Transformed program

Figure 15: Original and transformed programs

