# OpenReview forum: "LLM-Assisted Code Cleaning For Training Accurate Code Generators"
_ICLR.cc/2024/Conference — ICLR 2024 poster_

### Official Review · Reviewer_DhCg · 2023-10-30

**Soundness:** 3 good
**Presentation:** 3 good
**Contribution:** 3 good
**Rating:** 5
**Confidence:** 3

**Summary:**

This paper investigates data quality for code generation and finds that making the code more structured and readable leads to improved code generation performance. The authors build a data-cleaning pipeline to transform existing programs by 1.) renaming variables, 2.) modularizing and decomposing complex code into smaller helper sub-functions, and 3.) inserting natural-language based planning annotations. Experiments on two algorithmic code generation benchmarks indicate that fine-tuning on the transformed programs improves the code generation performance compared to fine-tuning on the original dataset.

**Strengths:**

. A nice idea to enhance code quality for code generation.

. A series of experiments were carried out to evaluate the effectiveness of the proposed method.

. The paper is easy to follow.

**Weaknesses:**

1. The proposed code transformations (i.e., renaming, modularizaing, and annotations) are a bit simple, and rely heavily on the capability of ChatGPT (GPT-3.5-TURBO).  It is unknown whether the proposed method can be utilized for other, more powerful LLM-based code generation systems. For example, in Section 4.2.2, the authors acknowledge that the poor performance obtained on the planning dataset may stem from the model's inability to generate accurate annotations, indicating that the effectiveness of the proposed model depends on LLMs.
2. The authors propose two steps of data cleaning (i.e., renaming, modularizaing) to the original source code. However, the authors did not validate the quality of the transformed code (the authors only tested whether the transformed code can be consistent to the original data). For example, whether the variable names actually became clearer and more readable after cleaning, and whether the model accurately segmented the code into modules. Hence, it is unknown whether such simple transformation process can enhance the quality of the training data.
3. As observed from the experimental results, the improvement of the proposed method could be insignificant and inconsistent. Some negative instances can occasionally be observed such as the $CL-7B + D_{modular}$ in in-context learning and the $CL-7B + D_{rename}$ in fine-tuning on the CODE-CONTESTS dataset. Unfortunately, the authors did not provide an explanation for the decline in these results, which may undermine the method's validity.
4. Code generation could be achieved by prompting a general LLM such as ChatGPT directly as well. Also, it has been found that by improving the prompts, the code generation performance can be improved. The authors may discuss this approach to accurate code generation: Liu et al., Improving ChatGPT Prompt for Code Generation, https://arxiv.org/abs/2305.08360

**Questions:**

. How is the quality of the transformed code?

. Does the effectiveness of the proposed model depend on LLMs such as ChatGPT?

---

> ### Author Response · Authors · 2023-11-18
> **Response to reviewer DhCg - Part I**
>
> We thank the reviewer for the constructive comments and feedback.
>
> > The proposed code transformations … rely heavily on the capability of ChatGPT (GPT-3.5-TURBO). It is unknown whether the proposed method can be utilized for other, more powerful LLM-based code generation systems.
>
>
> We acknowledge the reviewer's point that the efficacy of our code transformation approach is influenced by the underlying language model employed. We selected GPT-3.5-Turbo for its balance between transformation accuracy and operational cost. To substantiate the versatility of our method, we have conducted an additional ablation study by substituting GPT-3.5-Turbo with the more recent GPT-4-Turbo model. Owing to the significantly higher cost of GPT-4-Turbo (approximately 8-10 times higher), we use a subset of the original dataset and refer to the modularized version as D_Modularize_. To ensure comparability, we analyzed the same subset of original and transformed programs for both the base and modularized datasets. Further details on the experimental setup can be found in Appendix Section C.2.
>
>
> The table below illustrates that the D_Modularize_4 dataset yields improved performance over the D_Modularize dataset presented in the main paper. This enhancement not only demonstrates the adaptability but also the potential improvements of our approach using stronger models.
>
> |                           | Pass@1   | Pass@10  | Pass@25 |
> |---------------------------|----------|----------|---------|
> | CL-7B+Base                | 16.3     | 31.6     | 37.6    |
> | CL-7B+D_Modularize        | 18.8     | 33.0     | 38.2    |
> | CL-7B+D_Modularize_4      | **19.4**   | **34.3**   | **40.0**  |
>
>
>
> > The authors propose two steps of data cleaning (i.e., renaming, and modularizing) to the original source code. However, the authors did not validate the quality of the transformed code (the authors only tested whether the transformed code can be consistent to the original data)....
>
> We appreciate the reviewer's emphasis on the importance of quantifying code quality. We have conducted a thorough analysis of the transformed dataset, as detailed in Section 4.1, where we offer both qualitative and quantitative evidence of quality enhancement. Below, we summarize the insights from Section 4.1 that pertain to code *quality*:
>
> - Post-transformation, the median number of new functions introduced by the modularization process is three. This indicates that the modularization procedure actively decomposes existing programs to insert new, potentially more coherent functions.
> - The helper functions created during modularization typically embody standard algorithms frequently encountered in competitive programming, such as `dfs`, `build_graph`, `gcd`, `dp`, and `binary_search`. This suggests that the new helper functions are meaningful and positively contribute to the overall code.
> - A qualitative analysis of approximately 100 programs, including successful and unsuccessful transformations is presented with various success and failure modes detailed.
>
> Moreover, we posit that the quality of the dataset can be indirectly inferred from the improved performance of models trained on it. The performance gains achieved through our modularization approach satisfy this measure of quality.
>
> Based on reviewers’ concerns, we have additionally performed an analysis of the generated dataset using GPT-4 as a judge, a method gaining traction for assessing outputs from free-form language models [1,2]. While this approach might have subtle limitations, it still offers valuable insights to address the concerns raised. We tasked GPT-4 to quantitatively assess the transformed programs in regards to the meaningfulness of the transformations and consistency of original and transformed programs (i.e. following the same program logic and structure). Specifically, we asked GPT-4 to rate the transformation on
>
> - variable names (rating between 1-3 corresponding to not helpful, somewhat helpful, very helpful)
> - function decomposition (rating between 1-3 corresponding to not helpful, somewhat helpful, very helpful)
> - program consistency (rating between 0-1 corresponding to inconsistent and consistent)
>
> Appendix C.1 provides prompts used for GPT-4 jude.  Analysis of about 1000 programs reveals that GPT-4 regards 99% of the transformations as helpful (of which only 3-5% of examples can judged as *can do better*). Similarly, over 99% of the transformed programs are judged as consistent with the original program. The following table provides a distribution of scores annotated
>
> | | Score distribution | Average |
> |---------------------------|----------|----------|
> |Variable names | {3 : 967, 2 : 28, 1 : 3} | 2.96 |
> | Function decomposition | {3 : 938, 2 : 59, 1 : 1} | 2.93 |
> | Consistency | {1 : 994, 0 : 4} | 0.994 |
>
> Appendix C.1 provides a qualitative analysis of GPT-4 scores and more details.
>
> Finally, we will release our generated dataset for further analysis for the community.

---

> ### Author Response · Authors · 2023-11-18
> **Response to reviewer DhCg - Part II**
>
> > As observed from the experimental results, the improvement of the proposed method could be insignificant and inconsistent.
>
> Modest Improvements - While the improvements appear modest, this is in large part also an artifact of the algorithmic code generation task being a challenging domain. Following, we provide some context for the two benchmarks:
>
> - APPS. Prior works using the APPS dataset, CodeRL[1] and PPOCoder[2] have reported a similar range of performance improvements – pass@1 improvement from 5.6 to 7.1 by CodeRL and 3.6 to 5.2 by PPOCoder.
>
> - CodeContests.  CodeContests dataset is an even more challenging dataset in comparison to APPS and thus is also harder to improve upon. For context, within the AlphaCode models, the 41B model does not exhibit notable improvements over the 9B model. Similarly, even GPT-3.5-Turbo achieves a pass@100 of 18, compared to 12 with our best-performing CodeLLaMa7B model. To our knowledge, the only other study that has tackled this dataset without closed-source API-gated models is “outline then generate” [5]. They proposed a tokenization approach and showed similar improvements against the AlphaCode baseline
>
> Inconsistent Improvements - We observe small performance drops on the CodeContests D_rename experiment. While we cannot ascertain the exact reason, we have the following hypotheses.
>
> Hypothesis 1 - Training examples in the CodeContests datasets on average already consist of good or standard variable naming practices and further renaming does not help (as also evidenced to a lesser degree for the APPS dataset)
> Hypothesis 2 - Variable names have a lesser impact on code generation capability and the primary impact comes from modularization, our key transformation.
>
> > Code generation could be achieved by prompting a general LLM such as ChatGPT directly as well.
>
> We thank the reviewer for suggesting the related work. We have included more detailed related work in code generation with LLMs now.
>
> However, we would like to re-emphasize that in this work our goal is to study and propose methods for improving data quality for code generation. Data quality is an important aspect of training LLMs. We hope our proposed approach and findings will be useful for improving the training process for future (code) language models.
>
> [1] Judging LLM-as-a-judge with MT-Bench and Chatbot Arena. Neurips datasets 2023
>
> [2] Large Language Models Are State-of-the-Art Evaluators of Code Generation. Arxiv 2023
>
> [3] Coderl: Mastering code generation through pretrained models and deep reinforcement learning. Neuips 2022
>
> [4] Execution-based code generation using deep reinforcement learning. TMLR 2023
>
> [5] Outline, then details: Syntactically guided coarse-to-fine code generation. ICML 2023

---

> ### Author Response · Authors · 2023-11-21
> **Gentle reminder for the rebuttal**
>
> Dear Reviewer DhCg,
>
> Thank you for your time in reviewing our paper and providing insightful and constructive comments. This is just a gentle reminder to revisit our responses and reply as to whether our response and clarifications have addressed the issues raised in the review. If you have any further questions, please do not hesitate to let us know.

---

### Official Review · Reviewer_FtQG · 2023-11-01

**Soundness:** 3 good
**Presentation:** 2 fair
**Contribution:** 3 good
**Rating:** 8
**Confidence:** 5

**Summary:**

The paper shows that improving the “quality” of a code dataset can improve the performance of a CodeLlama7B model fine-tuned on that dataset. Specifically, for every source file in a dataset, the authors propose to use a __second__ instruction-tuned language model (gpt-3.5-turbo) to perform three types of transformations in order:
1) rename variables to have semantically meaningful names,
2) “modularize” the code by breaking up large chunks of code into smaller functions,
3) prepend a “plan” before the code that summarizes the role of each individual function.

The authors use their synthetic dataset to a) provide few-shot examples to CodeLlama7B for in-context learning, b) fine-tune CodeLlama7B. Through this process, they show modest improvements in pass@k on the APPS and CodeContest data sets.

**Strengths:**

- The proposed idea is simple and an interesting approach to re-format data using a neural approach. Especially given the fact that the domain is code, transformations can be verified using an oracle - which in this case is a set of test cases.

- The paper validates that by fine-tuning an LLM on a smaller good quality dataset, it is possible to achieve better/equal results than fine-tuning on a larger lower quality dataset - something that has been pointed out by many other papers in different context.

- The approach can be used as inspiration for other domains where the LLM is capable of editing an existing solution but incapable of generating an entirely new solution. However, the efficacy of this approach might be significantly impacted by the presence/absence of an oracle.

**Weaknesses:**

- The experiment section of the paper reports numbers on the subsets of two datasets. It would be nice to clearly outline the filtering criteria that are used for each dataset. I have certain questions regarding this in the “Clarifications section”.

- In many of the tables, there has been no reference or explanation to numbers that show a negative effect on results. For example, on the Code-Contest dataset, D_rename works worse than the baseline. It would be nice if the authors could be candid about this in their writing.

- Additionally, I believe that the CL-7B + D_distill number is missing in table 4(a). Can you please include that number in the rebuttal ?

- Overall, the effect of planning information is a mixed bag, and the conclusions are slightly confusing. For example, from the sentence, “Upon inspection of the generated solutions, we find that often the generated plans are imprecise or incorrect, highlighting that planning still remains a bottleneck.” – I am confused by this statement because I am unsure if this is because of the drawbacks of CodeLlama or a drawback of the paper’s approach. Overall I am unsure if D_planning actually supports the paper’s claim.

- The improvements on Code-Contests seem to be not as effective as would be expected.

**Questions:**

1. Comments on Fig 1:
    - In the renaming step, the variable `n` is not renamed everywhere. I understand that the actual LLM output can have mistakes, but maybe for an explanatory diagram this could be avoided.
    - Instead of “a -> root_u,  b -> root_v, …” as the text above the arrow, it would be clearer to show the natural language instruction that you provided (“Rename the variables in the program to be…”). As it stands currently, it looks like the renaming (“a -> root_u,  b -> root_v, …”) is the __input__ to the model. Same comment for the modularization and planning steps too.

2.  It’s not immediately obvious how the natural language “plans” are used (my initial understanding was that they are provided as a docstring for each function). Would be nice to clarify in Fig 1 that they are __prepended__ to the program as a comment.

3. In the APPs benchmark, do you consider all problems from “codeforces”, “codechef” and “atcoder” ? Or is there some further filtering done after that ? If further filtering has been done, can you please clarify what procedure has been followed?

4. What does this line “These cases are sometimes due to incorrect programs but more …. while only a single test solution is provided” mean ?

---

> ### Author Response · Authors · 2023-11-18
> **Response to reviewer FtQG**
>
> We thank the reviewer for the constructive comments and feedback.
>
> > However, the efficacy of this approach might be significantly impacted by the presence/absence of an oracle
>
> We agree with the reviewer that the equivalence-checking oracle plays a crucial role in maintaining the *data quality*. However, our approach can also accommodate learning-based "oracles" to eliminate poor-quality data points. For instance, [1] developed a quality classifier (through the manual annotation of a small data subset) for filtering GPT-3 outputs, while [2] utilized an entailment model to estimate the quality of text summarization pairs.
>
> > … clearly outline the filtering criteria that are used for each dataset…
>
> No additional filtering was performed for APPS. Further details are added to the appendix
>
> > In many of the tables, there has been no reference or explanation to numbers that show a negative effect on results.
>
> We thank the reviewer for the feedback and have updated the manuscript to reflect this.
>
> > Overall, the effect of planning information is a mixed bag, and the conclusions are slightly confusing .....
>
> We agree with the reviewer that results on D_planning do not provide consistent improvements and acknowledge the confusion. We will further clarify the claims about how planning affects performance.
>
> To present our findings and conclusion in perspective, we would like to highlight that improving the *reasoning* and *planning* capabilities is an active area of research [3]. Prior works have reported strong performance improvements by using a supervised learning paradigm by training on natural language plans or reasoning chains [4,5,6] albeit typically in less complex domains. In line with these approaches, we adopted the supervised learning paradigm, aiming to refine planning and reasoning for models by training on automatically curated natural language plans using our methodology. We do not assert that supervised learning is the definitive solution here, as discussed in Section 4.1 (final paragraph).
>
> While the results from D_planning are mixed, we believe they raise critical questions for future research. Two particularly promising avenues are:
>
> - Defining "quality" for natural language plans: The mixed success with planning might be attributed to inaccuracies in the automatically generated plans. Future data curation techniques that filter or augment this precision (akin to entailment models used in [2]) are promising.
>
> - Alternative learning paradigms for enhancing planning: The supervised learning paradigm followed in this work might be insufficient for models to generalize planning in complex domains. Future work can explore alternative learning algorithms, possibly over our modularization approach which naturally decomposes programs usually beneficial for solving complex problems.
>
> We have revised the manuscript to reflect this message more clearly.
>
> > The improvements on Code-Contests seem to be not as effective as would be expected.
>
> While CodeContests improvements do not seem significant in magnitude, we want to emphasize that the CodeContests dataset is an even more challenging dataset in comparison to APPS and thus also harder to improve upon. For context, within the AlphaCode models, the 41B model does not improve notably over the 9B model, suggesting that even scaling has a limited effect. To our knowledge, the only other study that has tackled this dataset without closed-source API-gated models is “outline-generate” [7]. They proposed a tokenization approach and observed similar improvements against AlphaCode.
>
> > What does this line “These cases are sometimes due to incorrect programs but more …. while only a single test solution is provided” mean?
>
> This refers to the observation that certain problems admit multiple correct solutions that are not equivalent to one another. For instance, a problem may allow the output of a list of numbers in any order. Our examination of the datasets revealed that both APPS and CodeContests typically include test cases that cover only one of the many correct solutions (e.g., only one specific ordering of the list). Example - https://codeforces.com/problemset/problem/1512/B.
>
> > Minor issues - Figure 1, missing distillation number, prepending plans
>
> Fixed and updated in manuscript.
>
> [1] Symbolic knowledge distillation: from general language models to commonsense models. ACL 2021
>
> [2] Referee: Reference-Free Sentence Summarization with Sharper Controllability through Symbolic Knowledge Distillation. 2022
>
> [3] Towards reasoning in large language models: A survey. 2023.
>
> [4] Distilling multi-step reasoning capabilities of large language models into smaller models via semantic decompositions. 2022
>
> [5] Symbolic Chain-of-Thought Distillation: Small Models Can Also" Think" Step-by-Step. 2023
>
> [6] Orca: Progressive learning from complex explanation traces of gpt-4. 2023
>
> [7] Outline, then details: Syntactically guided coarse-to-fine code generation. ICML 2023

---

> > ### Comment · Reviewer_FtQG · 2023-11-22
> >
> > Thank you for answering all my questions. The planning section reads much better now. I will update my score.

---

> ### Author Response · Authors · 2023-11-21
> **Gentle reminder for the rebuttal**
>
> Dear Reviewer FtQG,
>
> Thank you for your time in reviewing our paper and providing insightful and constructive comments. This is just a gentle reminder to revisit our responses and reply as to whether our response and clarifications have addressed the issues raised in the review. If you have any further questions, please do not hesitate to let us know.

---

### Official Review · Reviewer_MNi4 · 2023-11-03

**Soundness:** 3 good
**Presentation:** 4 excellent
**Contribution:** 3 good
**Rating:** 8
**Confidence:** 3

**Summary:**

This work explores applying a proposed data-cleaning pipeline (1: renaming variables, 2: refactoring into helper functions, and 3: inserting natural language comments to guide generation) to two major datasets (APPS and CodeContests). Fine-tuning a CodeLLAMA 7B model on these cleaned datasets dramatically improves fine tuning efficiency (requiring 8x less data to match the same performance) and improves accuracy a decent amount (often 1.2x-1.3x). They don't find the 3rd form of refactoring (introducing planning-based comments) to yield improvements, and through an additional experiment they narrow this down to be due to the inability of the model to generate good plans (as opposed to its ability to follow the plans).

**Strengths:**

- This is a beautifully written paper, quite easy to follow and at just the right level of detail.
 - APPS and CodeContests are two very standard datasets so those were a great choice.
 - The experiment in Table 4b around ground truth plans is a very nice little experiment, I appreciate the inclusion of that
- The data efficiency results in Figure 3 are quite good, showing 8x less finetuning data is needed to achieve the same Pass@1 when finetuning on the refactored programs.
- I appreciate the smaller scale observations/insights on LLM prompting, which I think are generally nice thing to include in these sorts of conference papers for the community, e.g. "Finally, in accordance with existing literature on prompting LLMS, we found that using simple and precise, low-level instructions improves the performance and accu- racy of the models in performing the operatons. Thus, for complex data cleaning operations, we find improvements by breaking it down and performing multiple operations iteratively."
 - The main results (Table 3 and 4a) are decent (not incredible, but reasonable in my opinion).

**Weaknesses:**

- The improvements in Table 3 are okay, not huge but still noticeable.
- The planning results are also modest, but this is interesting in its own right, and the analysis of how ground truth plans would help considerably is a good way to isolate much of the problem to the plan creation rather than plan execution.
- While most of the paper was easy to read, I was quite unclear on the distillation dataset baseline – see the Questions section for details
- For more minor / easily fixed weaknesses see Questions section

**Questions:**

- In table 3 theres one missing entry – Pass@1 APPS Interview Distill. Where is it?

- I don't understand the "distill" baseline dataset laid out at the end of 3.2, and referenced at various points
    - My best guess is that you're doing synthetic data generation to generate a new dataset by prompting with few-shot examples from the modular dataset? Is it generating both the test cases *and* the solutions to them? Or are the test cases taken from somewhere and then its just generating solutions?

- Two relevant pieces of work on synthetic data generation of code for LLMs are the Self Taught Reasoner (STaR) (Zelikman et al 2022) and Language Models Can Teach Themselves to Program Better (Haluptzok et al 2022). Those would be relevant to reference under the "Synthetic data for LLMS" section of Related Work, and could also relate to the distillation (though as mentioned before, I understand the distill baseline less).

Low level confusing things:
- Given that the 30% relative improvement is in Table 4, it's confusing that the caption of Table 3 brings up the 30% statistic (led to me spending a while trying to figure out which two numbers divide to get 30%, which is none in table 3)
- Note: missing period in last paragraph of Section 1 right before "Next"
- Typo at end of 2.1 with random sentence ending: "steps quite effectively. effective in generating high-quality outputs."
- The sentence "We obtain three *parallel* datasets at the end of our cleaning process, one for each of renaming, modularization, and planning" and in particular the world "parallel" is a bit misleading since at least to me it suggests that each dataset comes from applying a single transformation to the original dataset independent of the others, but actually the 3 transformations build on each other. This is clarified by Table 2 but would be helpful to have in the text as well.
- Table 4 isn't actually labelled "Table 4" anywhere (since there's no shared caption)

---

> ### Author Response · Authors · 2023-11-18
> **Response to reviewer MNi4**
>
> We thank the reviewer for the constructive comments.
>
> > The improvements in Table 3 are okay, not huge but still noticeable.
>
> While the improvements in Table 3 do not appear to be huge, this is in large part also an artifact of the algorithmic code generation task being a challenging domain. For example, prior works using the APPS dataset, CodeRL[1] and PPOCoder[2] have reported a similar range of performance improvements – pass@1 improvement from 5.6 to 7.1 by CodeRL and 3.6 to 5.2 by PPOCoder. Finally, we study the data quality for improving code LLMs which is an important problem and complements existing works. Similarly, CodeContests is even more challenging and harder to improve upon.
>
> > The planning results are also modest, but this is interesting in its own right, and the analysis of how ground truth plans would help considerably is a good way to isolate much of the problem to the plan creation rather than plan execution.
>
> We agree with the reviewer that planning results are modest/mixed bag. To add additional context and perspective – improving the reasoning and planning capabilities of language models is a challenging problem and a research direction actively explored by the community [10]. Our experiment of disentangling plan generation with code implementation shows that even after training on such plans, current models cannot generalize and construct plans for new challenging problems. To provide the insights more clearly, we have added some additional context in the paper summarized below:
>
> We adopted the supervised learning paradigm, aiming to refine planning by training on natural language plans curated using our methodology akin to prior works [4,5,6]. While the results from D_planning are mixed, we believe they raise critical questions for future research. Two particularly promising avenues are:
>
> - Defining "quality" for natural language plans: The mixed success with planning might be attributed to inaccuracies in the automatically generated plans. Future data curation techniques that filter or augment this precision (akin to entailment models used in [2]) is promising.
>
> - Alternative learning paradigms for enhancing planning: The supervised learning paradigm followed in this work might be insufficient for models to generalize planning in complex domains. Future work can explore alternative learning algorithms, possibly over our modularization approach which naturally decomposes programs usually beneficial for solving complex problems.
>
>
> > Clarifications on the distillation experiment
>
> Yes, in the distillation experiment, we take the problem statements from the training dataset and use the GPT-3.5-Turbo model to generate multiple, *diverse* solutions using a few-shot prompting approach. Next, we filter the generated solutions for correctness based on the ground truth test cases provided in the dataset to ensure we are not training on incorrect hallucinated solutions. This experiment is meant to provide a strong baseline that distills larger-model-generated outputs which has proven to be a very popular approach for both instruction tuning and task-specific finetuning [6,8,9].
> In contrast to distillation, our transformation approach enjoys the benefits of performing the simpler operation of transforming (editing) existing solutions instead of directly generating them. Thus it yields about 94\% successful transformation rate whilst the distillation baseline can only generate solutions for about 50\% of the problems.
>
> > Two relevant pieces of work on synthetic data generation …
>
> Yes, these works are relevant. STaR learns natural language reasoning (corresponding to our planning approach) in simpler domains. Haluptzok et al generates datasets using language models and further models on them. We have added them to the related works.
>
> > Smaller issues – missing distillation number, typos, usage of the word parallel, table 4 caption
>
> We have fixed these issues and updated the paper.
>
> [1] Coderl: Mastering code generation through pretrained models and deep reinforcement learning. Neuips 2022
>
> [2] Execution-based code generation using deep reinforcement learning. TMLR 2023
>
> [3] Outline, then details: Syntactically guided coarse-to-fine code generation. ICML 2023
>
> [4] Distilling multi-step reasoning capabilities of large language models into smaller models via semantic decompositions. Arxiv 2022
>
> [5] Symbolic Chain-of-Thought Distillation: Small Models Can Also" Think" Step-by-Step. Arxiv 2023
>
> [6] Orca: Progressive learning from complex explanation traces of gpt-4. Arxiv 2023
>
> [7] Referee: Reference-Free Sentence Summarization with Sharper Controllability through Symbolic Knowledge Distillation. 2022
>
> [8] Wizardlm: Empowering large language models to follow complex instructions. Arxiv 2023
>
> [9] WizardCoder: Empowering Code Large Language Models with Evol-Instruct. Arxiv 2023
>
> [10] Towards reasoning in large language models: A survey. Arxiv 2023

---

> ### Author Response · Authors · 2023-11-21
> **Gentle reminder for the rebuttal**
>
> Dear Reviewer MNi4,
>
> Thank you for your time in reviewing our paper and providing insightful and constructive comments. This is just a gentle reminder to revisit our responses and reply as to whether our response and clarifications have addressed the issues raised in the review. If you have any further questions, please do not hesitate to let us know.

---

> > ### Comment · Reviewer_MNi4 · 2023-11-21
> >
> > Thank you for the discussion in the rebuttal and for adding clarification to the text discussion around distillation – having gone through the paper again, I've adjusted my score to an 8.

---

### Author Response · Authors · 2023-11-18
**Summary of the changes**

We thank all the reviewers for providing constructive feedback for the paper. We have updated the manuscript clarifying aspects of the approach, results, and conclusions. In addition, we have performed additional ablation and evaluations demonstrating the generality of our approach. Following, we list down the revisions made to the paper (in blue ink) -

- **GPT-4 as a judge evaluation of transformations**. We use GPT-4 to provide further insights into the quality of transformed programs. We use GPT-4 as a judge, a method gaining traction for assessing outputs from free-form language models [1,2]. While this approach might have subtle limitations, it still offers valuable insights to address the concerns raised. We tasked GPT-4 to quantitatively assess the transformed programs in regards to the meaningfulness of the transformations and consistency of original and transformed programs (i.e. following the same program logic and structure). Appendix C.1 provides a more comprehensive experimental setup with prompts used for GPT-4. Analysis of about 1000 programs reveals that GPT-4 regards 99% of the transformations as helpful (of which only 3-5% of examples are judged as can do better). Similarly, over 99% of the transformed programs are judged as consistent with the original program. See Appendix C.1 and Section 4.1 ending for further details.
- **Ablation on the choice of model for performing the transformations**. We selected GPT-3.5-Turbo for its balance between transformation accuracy and operational cost. To substantiate the versatility of our method, we have conducted an additional ablation study by substituting GPT-3.5-Turbo with the more recent GPT-4-Turbo model. We generate modularized programs using GPT-4-Turbo and compare them against the *parallel* transformed programs generated using GPT-3.5-Turbo. Results demonstrate that not only we can perform transformations using GPT-4-Turbo but it also provides performance improvements owing to its better instruction following capacity. See Appendix C.3 and Section 4.2.3 end for further details
- **Clarifying the results and conclusion from the planning experiment**. We have provided more context about our choice of supervised learning algorithm for improving planning and future research directions motivated by our approach. See Section 4.2.2 for the update.
- **Updates to conclusion**. We have provided more granular future work highlighting tasks where fixed oracles are absent (using "learned" oracles analogous to [3,4]) and other software engineering tasks. See Section 6 for the update.
- **Additional related work for synthetic data and code llms**. Updated section 5 and added Appendix D.
- **Minor updates**. Fixed typos, figure and table captions, modified Figure 1 to be more coherent and added the missing number for D_distill.

[1] Judging LLM-as-a-judge with MT-Bench and Chatbot Arena. Neurips datasets 2023

[2] Large Language Models Are State-of-the-Art Evaluators of Code Generation. Arxiv 2023

[3] Symbolic knowledge distillation: from general language models to commonsense models. ACL 2021

[4] Referee: Reference-Free Sentence Summarization with Sharper Controllability through Symbolic Knowledge Distillation. 2022

---

### Meta-Review · Area_Chair_2n5A · 2023-12-05

**Metareview:**

This paper presents a method to improve the code generation skills of large language models by fine-tuning models on cleaned code data in higher quality. In particular, such high-quality code data is generated by a data cleaning pipeline that revises existing code datasets by applying three transformations: (1) renaming variables with idiomatic and semantically meaningful names, (2) modularizing programs by decomposing complex logics into individual functions, (3) augmenting modularized code with step-wise natural language plans (e.g., explanations). These transformations are implemented by prompting GPT-3.5-turbo. The authors experimented on two challenging code generation benchmarks: APPS and CodeContests, showing modest to strong improvements on CodeLlama-7B against a fine-tuning baseline using original training data, also outperforming a “self-training”-styled data augmentation approach (denoted as distillation in the paper) that generates new code solutions by solving code problems from scratch instead of revising existing solutions in the training sets. This approach is more data efficient, outperforming the fine-tuning baseline by using just ⅛ of revised code data. Most performance improvements are from using variable renaming and code modularization, while the step-wise natural language explanation yields mixed results. Finally, the authors further explain that the mixed results on natural language planning steps was largely due to data quality issues, while the model is able to generate correct programs given ground-truth natural language steps.

**Strengths:**

The paper is clearly written and easy to follow (MNi4, MNi4)

The reviewers agreed that improving code data quality by revising existing code corpora is a pretty neat idea (FtQG, DhCg) and validates the common observation that fine-tuning LLMs on smaller, high-quality datasets is better than using larger but noisier datasets (FtQG). The authors also presented a data efficiency analysis that offers further insights (MNi4).

The improvements on challenging program synthesis benchmarks (APPS, CodeContest) are quite promising in order to scale this approach up for large code language models or apply this to other domains (FtQG). I appreciate that the authors attempted those harder benchmarks instead of classical datasets such as MBPP/HumanEval.

**Weaknesses**

Resolved Issues: all the reviewers agreed that two major weaknesses of the paper are:

(1) Improvements on some splits of the benchmarks (APPS-INTERVIEW, CodeContests) are modest (0.8% ~ 1.9% absolute vs fine-tuning on original training data). This issue was mostly resolved through the author discussion phase, as the authors pointed out that related works observed similar gains on these two challenging benchmarks.

(2) Natural language step-wise planning yields mixed results, and the initial version of the paper didn’t clearly discuss those results. The authors provided more clarifications and explanations to the mixed results in the updated version, while highlighting that the general theme of improving task performance via natural language reasoning is still an open research question, which is recognized by reviewers MNi4 and FtQG.

Open Issues: Besides, reviewer DhCg also raised two valid concerns: lacking intrinsic evaluation of the quality of revised code **(open-issue-1)**, and lacking evidence to support that this approach would work for more capable code LLMs other than CodeLlama-7B **(open-issue-2)**.

The authors responded to open-issue-1 by auto-rating the quality of revised code using GPT-4, showing very positive results. While using GPT-4 to rate responses from weaker models is a common practice, it is unclear whether such ratings are aligned with human judgment. The authors further showed a qualitative analysis, which seems convincing, but a manual annotation of code quality rate could be more favorable.

For open-issue-2, it seems the authors didn’t directly address this issue (their response was about using a stronger model to further improve code revision quality, as opposed to fine-tuning a stronger model using the revised code). Still, applying this approach to larger LLMs might be out of the scope of this paper.

While these two issues remain unresolved, they are not critical enough. The proposed data revision approach is intuitive and works reasonably well on challenging code generation tasks. Therefore, the decision is "Accept".

**Justification For Why Not Higher Score:**

As pointed out by reviewer FtQG, while the approach of improving code LLMs by reformatting its training data is an overlooked direction and the method can be used as inspiration for other domains where the LLM is capable of editing an existing solution but incapable of generating an entirely new solution. However, the efficacy of this approach might be significantly impacted by the presence/absence of an oracle (test cases to verify functional correctness of transformed solutions). Given that it's challenging to adapt this method to other applications, this may potentially limit the level of excitement received from a broader audience outside of program synthesis community.

**Justification For Why Not Lower Score:**

Most technical issues were resolved during the rebuttal phase, and the reviewers (2/3) are generally excited about this paper.

---

### Decision · Program_Chairs · 2024-01-16

Accept (poster)